# Continuous cholinergic-dopaminergic updating in the nucleus accumbens underlies approaches to reward-predicting cues

Miguel Skirzewski [1] ✉, Oren Princz-Lebel[1,2], Liliana German-Castelan [1,2], Alycia M. Crooks [1,2], Gerard Kyungwook Kim [1,6], Sophie Henke Tarnow[1,7], Amy Reichelt[1,8], Sara Memar[1], Daniel Palmer [1,3], Yulong Li [4], R. Jane Rylett[1,3], Lisa M. Saksida [1,3], Vania F. Prado [1,3,5], Marco A. M. Prado [1,3,5] ✉ & Timothy J. Bussey [1,3] ✉

The ability to learn Pavlovian associations from environmental cues predicting positive outcomes is critical for survival, motivating adaptive behaviours. This cued-motivated behaviour depends on the nucleus accumbens (NAc). NAc output activity mediated by spiny projecting neurons (SPNs) is regulated by dopamine, but also by cholinergic interneurons (CINs), which can release acetylcholine and glutamate via the activity of the vesicular acetylcholine transporter (VAChT) or the vesicular glutamate transporter (VGLUT3), respectively. Here we investigated behavioural and neurochemical changes in mice performing a touchscreen Pavlovian approach task by recording dopamine, acetylcholine, and calcium dynamics from D1- and D2-SPNs using fibre photometry in control, VAChT or VGLUT3 mutant mice to understand how these signals cooperate in the service of approach behaviours toward reward-predicting cues. We reveal that NAc acetylcholine-dopaminergic signalling is continuously updated to regulate striatal output underlying the acquisition of Pavlovian approach learning toward reward-predicting cues.

The ability to learn to associate environmental cues with positive outcomes is critical for survival. These Pavlovian associations directly enable the acquisition of complex emotional and motivational states toward cues signalling food sources (i.e. rewards)[1]. These reward-predicting cues can acquire motivational properties themselves, in the form of incentive salience, or 'wanting', which can manifest as seeking behaviours such as approaching the predictive positive environmental cue[2]. Importantly, substantial evidence from both human and animal studies supports the hypothesis that endophenotypes in addiction, schizophrenia, depression, Parkinson's disease, and other forms of psychopathological processes are the result of abnormal incentive salience processing[1].

Robust and well-established brain circuit mechanisms including the activity of dopamine (DA) in the nucleus accumbens (NAc) regulate incentive salience[3–7]. However, several studies have suggested an equally important role for NAc acetylcholine (ACh)[8–14]. ACh in the NAc

[1]Robarts Research Institute, Schulich School of Medicine & Dentistry, Western University, London, ON N6A5K8, Canada. [2]Neuroscience Program, Schulich School of Medicine & Dentistry, Western University, London, ON N6A5K8, Canada. [3]Department of Physiology and Pharmacology, Schulich School of Medicine & Dentistry, Western University, London, ON N6A5K8, Canada. [4]State Key Laboratory of Membrane Biology, Peking University School of Life Sciences, Beijing 100871, China. [5]Department of Anatomy and Cell Biology, Schulich School of Medicine & Dentistry, Western University, London, ON N6A5K8, Canada. [6]Present address: Centre for Biological Timing and Cognition, Department of Cell and Systems Biology, University of Toronto, Toronto, ON M5S3G3, Canada. [7]Present address: Cognitive Development and Neuroimaging Laboratory, Western University, London, ON N6A3K7, Canada. [8]Present address: School of BioMedicine, Faculty of Health and Medical Sciences, University of Adelaide, Adelaide, SA 5005, Australia. ✉e-mail: mskirzew@uwo.ca; mprado@uwo.ca; tbussey@uwo.ca

is released by tonically active cholinergic interneurons (CINs)[8,15–17], which modulate DA release via nicotinic receptors[18–21]. CINs also directly regulate, via M1- and M2-class muscarinic receptors[22–26], or indirectly via the activation of parvalbumin-positive GABAergic interneurons[26,27], the concurrent activity of D1- and D2-expressing spiny projecting neurons (SPNs) projecting to the direct striatonigral (D1-SPNs) or indirect striatopallidal (D2-SPN) basal ganglia pathways[28].

It has been suggested that NAc CINs encode motivational signals supporting approach or avoidance behaviours[29,30]. For example, microdialysis studies in rodents have shown that extracellular ACh levels in NAc increase during conditions that reduce reward-seeking behaviours such as satiety[31,32], conditioned taste aversion[33], anxiety-like and depression-like states[34,35], and drug or sugar-binge withdrawal[32,36–39]. In contrast, salient reward-predicting cues that encourage motivated behaviour have been shown to promote a characteristic 'pause' in CIN firing[8,9,13,40,41]. This CIN pause coincides with phasic DA activity during learning[11,13], suggesting a CIN-DA gating mechanism regulating plasticity at corticostriatal synapses onto SPNs[22,42].

Despite this emerging evidence for a role of both NAc DA and ACh in cue-motivated behaviours, little is known about the relationship and interaction between DA and CIN-mediated signalling during the acquisition of Pavlovian associations. Understanding this relationship in vivo is especially critical given the potential role of both DA and CINs in the need to constantly update cue-reward associations to optimise reward-mediated behaviours, and how its disruption contributes to neurodevelopmental, psychiatric, and neurodegenerative disorders[43–45].

Another unanswered question regarding NAc reward learning-related circuitry is whether the critical neurotransmitter released by CINs is, in fact, ACh. This is a relevant question because CINs have been shown to co-release glutamate, mediated by expression of the vesicular glutamate transporter 3 (VGLUT3)[46]. Indeed, activation of VGLUT3-mediated glutamate release from CINs can directly affect plasticity in SPNs[47,48], regulating DA release[49] and addiction-related behaviours[50]. A reason this question remains unanswered is that experiments aimed at manipulating or recording CIN activity typically involve local lesioning, optogenetics, chemogenetics, calcium imaging, pharmacology and/or electrophysiology, all of which have been highly informative, but none of which can distinguish the role of ACh and glutamate released from CINs.

To investigate the interactions between CIN-released neurotransmitters and NAc circuitry, we used genetically-encoded sensors and fibre photometry to record millisecond dynamics of ACh, DA, and calcium in putative D1- and D2-SPNs, in the NAc of mice during acquisition of a task that measures Pavlovian approach behaviours to reward-signalling cues[51]. We found that highly coordinated DA-ACh signalling underlies reward prediction and reward collection. Mice with disrupted striatal VGLUT3 behaved normally. However, decreased levels of the vesicular acetylcholine transporter (VAChT) in striatal CINs, which significantly reduces ACh release from CINs[49], abolished coordinated DA signalling and disrupted concurrent D1- and D2-SPN calcium activity and Pavlovian approach behaviours, which was rescued by restoring VAChT in the NAc. Our results reveal how balanced dopaminergic-cholinergic signalling in the NAc regulates striatal outputs in the service of updating cue-motivated learning in mice.

## Results
### Mice performing the Autoshaping task exhibit approach behaviours directed towards stimuli predicting rewards

The Autoshaping task (Fig. 1) is a well-established Pavlovian behavioural paradigm for rodents that assesses the motivational and incentive salience properties of a rewarding unconditioned stimulus (US) and a neutral conditioned stimulus (CS) predicting rewards[51–62]. Briefly, repeated paired presentations of a CS anticipating rewards (CS+) can elicit conditioned responses including approaches toward the CS+, even though no response from the animal is required. This phenomenon is often referred to as sign-tracking. Presentation of a CS that is not associated with reward (CS-) leads to a decrease in approaches toward the CS-. Another type of conditioned response often observed in rodents is the development of approach behaviours toward the location of the US delivery (reward magazine) during the CS presentation, despite the rewards not being delivered until after the termination of the CS+[63]. This phenomenon is often referred to as goal-tracking. We initially studied the behaviour of wild-type C57BL/6j mice (n=12♂, n=12♀) using the touchscreen-based Autoshaping task (Supplementary Fig. 1). Both male and female mice learned the association between the CS+ and delivery of a strawberry milkshake reward (10 µl), evidenced by an increase in the time mice spent approaching the CS+, and a reduction in the time spent approaching the CS- (S1→S10, Supp.Fig. 1a-d). When the reward contingency was reversed (S11→S20, Supp.Fig. 1a), both male and

**Fig. 1 | The touchscreen Autoshaping task to assess Pavlovian approach behaviours toward reward-predicting stimuli. a** Layout of the Autoshaping touchscreen operant chamber depicting the two screens (left, CS-; right, CS+) and the reward magazine (RM) delivering strawberry milkshake reward (10 µl). Each chamber was equipped with a back infrared photobeam (BIR) to initiate trials, and two front infrared photobeams (FIR) on each side of the RM to record approaches to the CS screen. An infrared photobeam inside the RM (not displayed) recorded latency time to collect rewards. **b** Flowchart overview of the Autoshaping task during acquisition (left) and reversal (right) training sessions. (left) Following a variable ITI, a trial initiated after breaking the BIR followed by the presentation of the stimulus (CS+ or CS−) during 10 s. Upon CS+ offset a reward was delivered and a new ITI began once the mouse pulled away from the RM. Upon CS− offset, no

reward was delivered, and a new ITI started. Within a single session, CS+ and CS− trials alternated pseudo-randomly. In total, each session ended after 20 CS+ and 20 CS- trials or after 60 min, whichever occurred first. (right) Following 10 acquisition sessions (1 session/day), mice undergo a total of 10 reversal sessions, in which the location of the CS+ and CS- were reversed. **c** (left) In contrast to the previous, both CS screens (left and right) had 50% of probability to deliver rewards in non-deterministic trials. Contingencies after CS+ or CS- remained similar as previously described. Within a single session a total of 20 CS+ and CS- trials were presented. (right) After 10 consecutive non-deterministic training sessions, mice followed 10 consecutive deterministic training sessions as described in (**b**). Figure 1a was created with BioRender.com.

female mice initially spent more time approaching the new CS- (former CS+), and then shifted after several sessions towards spending more time approaching the new CS+ (former CS-). No sex differences in approach behaviours to the CS were found ($p > 0.05$). The touchscreen-based Autoshaping task is designed to record approach behaviours toward the location of the CS. In the present study we also recorded nose-pokes to the reward magazine. We observed that both male and female mice showed little time nose-poking the reward magazine during the CS presentation (Supp.-Fig. 1e-f), indicating that the task set-up is effective in eliciting approach behaviours towards the location of the CS almost exclusively (i.e., sign-tracking). As there were no differences between male and female mice on approaches to CS+ or CS-, in subsequent experiments we combined males and females into a single group for analysis.

## Nucleus accumbens dopamine dynamics correlate with approach behaviours in the Autoshaping task

The formation of cue-reward associations depends on DA release in the NAc[59,64–66]. Here, we combined the recently developed GRAB$_{DA2m}$[67] biosensor (hereafter GRAB$_{DA}$) with fibre photometry to characterise in vivo extracellular NAc DA dynamics in wild-type C57BL/6j mice ($N = 8$, $n = 4\male$, $n = 4\female$) learning the Autoshaping task (Fig. 2a–c, Supp.Fig. 2a–e). In addition, we recorded DA in an independent cohort of C57BL/6j mice ($N = 9$, $n = 5\male$, $n = 4\female$) performing a 'non-deterministic' variation of the Autoshaping task in which, on a given trial, the location of the CS+/CS- was determined pseudo-randomly (50% probability) (Fig. 1c and Fig. 2a). In this version of the task mice were unable to predict which lit location (either left or right screen) was associated with reward delivery. After ten such sessions (S1→S10), mice underwent ten standard 'deterministic' sessions (S11→S20) in which the CS+/CS- location remained constant within and across sessions.

Using this combined approach, we obtained robust and reliable recordings of extracellular DA levels within the NAc (Fig. 2b and Supp. Fig. 2c) that progressively changed across training sessions (Fig. 2c–f). We found that mice tethered for fibre photometry recordings behaved similarly to control mice without fibre optical implants during the Autoshaping task ($p > 0.05$, Fig. 2a and Supp. Fig. 1a), indicating no major effect of tethering or surgical implants.

DA dynamics were tightly coupled to approaches toward CS presentation (Fig. 2g-left panel and Supp. Fig. 2d–g). Specifically, as mice learned the task during acquisition sessions (S1→S10), the amplitude of the DA response became consistently larger during presentation of the CS+ compared to presentation of the CS-. Such changes were not seen during non-deterministic contingencies. Interestingly, when the locations of the CS+ and CS- were first reversed (S11), a large increase in DA levels was observed during CS-presentation (former CS+) which did not change during the CS+ (former CS-). Finally, after five consecutive reversal sessions (S11→S15) the amplitude of DA response during stimulus presentation was larger during the CS+ compared to the CS- (Fig. 2g-left panel and Supp. Fig. 2f,g). Similarly, once mice performing non-deterministic contingencies began deterministic training contingencies (S11→S20), the DA response became significantly larger during CS+ trials.

Following CS+ offset, a phasic DA response was observed during reward delivery (Fig. 2g-right panel and Supp. Fig. 2h). Across the acquisition and reversal training sessions, the amplitude of this reward-evoked DA response progressively reduced as mice learned the association between CS+ and reward. In contrast, the amplitude of reward-evoked DA responses during the CS+ and CS- in non-deterministic contingencies remained constant across sessions (S1→S10), but when deterministic contingencies were established (S11→S20), the amplitude of the DA response significantly decreased across sessions.

DA dynamics during CS+, CS-, and reward collection were closely correlated with approaches during the Autoshaping task (Fig. 2h, i). During acquisition and reversal sessions, the longer mice spent approaching the CS+ compared to the CS-, the larger the relative increase of NAc DA signal during CS+ trials (Fig. 2h, top panels). No correlation was observed on non-deterministic sessions (Fig. 2h, bottom panels). Also, the time mice spent approaching the CS+ was inversely correlated with the amplitude of reward-evoked DA responses (Fig. 2i, top panels), although this effect was not significant in the group of mice trained first in the non-deterministic contingencies (Fig. 2i, bottom panels). Together, our findings indicate that NAc DA dynamics using the Autoshaping task strongly correlate with approach behaviours and reward predictability, as previously demonstrated in other paradigms[68–72].

## Acetylcholine release from striatal cholinergic interneurons regulates the acquisition of approach behaviours

Within the NAc, CINs are proposed to play fundamental roles in modulating presynaptic DA release[18–20], regulating the activity of local circuits[27,73], and integrating environmental information to regulate behaviour[8–14]. However, NAc CINs also co-release glutamate[46], and it remains unclear whether these functions depend mostly on ACh or glutamate release[74]. To disentangle the individual contributions from ACh or glutamate released from CINs in approach behaviours, we used two genetically modified mouse lines (VAChTcKO and VGLUT3cKO)[49,75] with selective knockout of the vesicular ACh transporter (VAChT, Fig. 3a) or the vesicular glutamate transporter (VGLUT3, Fig. 3b) in the striatum[46–48,75,76]. These two proteins are required for ACh or glutamate release from CINs, respectively. We found that VAChTcKO (VAChTcKO: $N = 25$, $n = 11\male$, $n = 14\female$; control: $N = 24$, $n = 13\male$, $n = 11\female$; Fig. 3c and Supp. Fig. 3), but not VGLUT3cKO mice (VGLUT3cKO: $N = 23$, $n = 12\male$, $n = 11\female$; control: $N = 24$, $n = 12\male$, $n = 12\female$; Fig. 3d and Supp. Fig. 4) failed to discriminate between the CS+ and CS- during acquisition sessions, demonstrated by their equal time spent approaching the CS+ and CS- during presentation (Fig. 3c and Supp. Fig. 3b). Interestingly, VAChTcKOs spent more time approaching the CS+ compared to the CS- during late reversal sessions, suggesting that some basic learning ability is preserved. This discrimination ability was not demonstrated until the ~17th session of training, indicating a severe learning impairment. This cannot be interpreted as intact reversal learning as these mice did not acquire the association initially, so for them there was no association to reverse. No sex differences were observed across genotypes when compared with their control littermates ($p > 0.05$).

Given this substantial behavioural impairment in VAChTcKOs, we next assessed whether in vivo ACh dynamics in the NAc changed during the acquisition of approach behaviours. We used GRAB$_{ACh3.0}$ (hereafter ACh3.0)[77] injected within the NAc of an independent cohort of mice (VAChTcKO: $N = 8$, $n = 4\male$, $n = 4\female$; control: $N = 7$, $n = 4\male$, $n = 3\female$) to record rapid dynamic changes of extracellular ACh using fibre photometry during performance of the Autoshaping task (Fig. 4a–d and Supp. Fig. 5a–d). During CS+ trials, we observed a significant decrease (~5-8s long) in ACh signalling across acquisition and reversal sessions after reward delivery in control littermate mice, but not in VAChTcKOs (Fig. 4b–e and Supp. Fig. 5e-g). Previous electrophysiological findings suggest that CIN pauses following a salient stimulus are critical to rapidly gate the influx of cortical inputs and synaptic plasticity onto SPNs to invigorate reward-predicting behaviours[8,9,13,40,41]. Consistently, our observations suggest that CIN-mediated pausing of tonic ACh release during rewards may underlie the development of approaches toward CS+ in mice. A previous report using microdialysis has shown that tonic striatal extracellular ACh levels in VAChTcKO are significantly reduced (~95%)[49], which may limit the ability to detect decreased cholinergic signals using ACh3.0. It is therefore likely that cholinergic tone in VAChTcKO mice is so low that

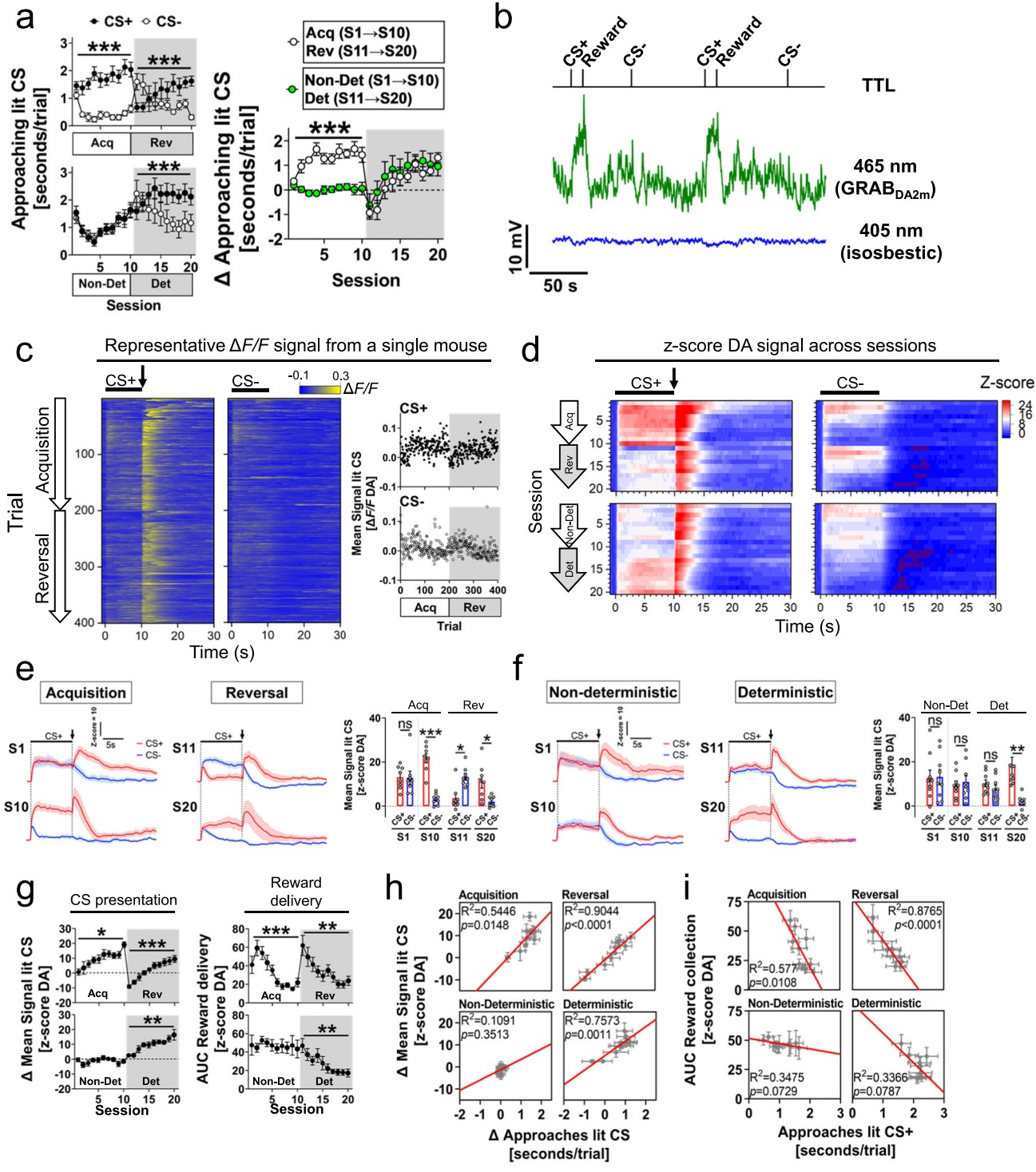

changes in CIN activity (such as pauses in activity) are unable to further modulate cholinergic tone. Additionally, a phasic increase (~1s) in ACh signal that did not differ between genotypes was observed during both CS+ and CS- onset ($p > 0.05$, Supp. Fig. 5h). We also observed a phasic ACh response during CS+ offset that was significantly impaired in VAChTcKOs (Supp. Fig. 5i). This event was not observed during CS- offset. We found a significant inverted relationship between ACh response and approaches to CS+ in control littermates (Fig. 4f) but not in VAChTcKO mice (Fig. 4g) during acquisition training sessions. These results suggest that in mice with low levels of VAChT, presynaptic-mediated plasticity mechanisms regulating ACh release are severely impaired, disrupting Pavlovian approach behaviours.

CINs provide the primary source of ACh in the NAc[78], but it has been reported that cholinergic neurons from the brainstem project to the striatum[79] to regulate local circuits underlying action strategies and cognitive flexibility[80]. Therefore, we investigated whether mice lacking 90% of VAChT expression from brainstem cholinergic neurons projecting to the striatum (En1-cre,VAChT[flox/flox])[81] might display deficits in the Autoshaping task. We found that both En1-cre,VAChT[flox/flox] ($N = 17$, $n = 9\male$, $n = 8\female$) and control littermate ($N = 18$, $n = 9\male$, $n = 9\female$) mice were able to learn to approach the CS+ across training sessions (Supp. Fig.6), suggesting that the release of ACh from brainstem neurons projecting to the striatum and other brain regions contributes little to Pavlovian approach behaviours. Taken together, our work

**Fig. 2 | In vivo nucleus accumbens dopamine dynamics correlate with approach behaviours towards reward-predicting stimuli. a** (Top-left) Tethered C57BL/6j mice performing the Autoshaping task during acquisition (Acq) and reversal (Rev) sessions were compared with mice (bottom-left) performing non-deterministic (Non-Det) and deterministic (Det) sessions. In contrast to non-deterministic sessions (S1→S10), mice spent more time approaching the CS+ than the CS- during acquisition sessions (two-way RM-ANOVA SessionXCS interaction, Acq: $F(9,126) = 3.892$, $p = 0.0002$; Non-Det: $p > 0.05$). During reversal and deterministic sessions (S11→S20), mice spent more time approaching the CS+ than the CS- (two-way RM-ANOVA SessionXCS interaction, Rev: $F(9,126) = 7.205$, $p < 0.0001$; Det: $F(9,144) = 4.937$, $p < 0.0001$). (Right) The relative time ($\Delta$ [CS+ − CS-]) mice approached the CS demonstrated that during acquisition (blank circles) but non-deterministic sessions (green circles), mice approached the CS+ (Mixed-effects model SessionXCS interaction, Acq: $F(9,133)=3.362$, $p < 0.0001$; Non-Det: $p > 0.05$). Mice approached the CS+ during reversal and deterministic sessions ($p > 0.05$). **b** Representative fibre photometry raw fluorescence signal at 465nm (green, $\text{GRAB}_{DA2m}$) and isosbestic 405nm (blue). A TTL input signal time-locked CS onset and reward delivery. **c** Trial-by-trial DA heatmaps (left, CS+; right, CS-) from a representative mouse ($\Delta F/F$) performing the Autoshaping task. (Session 1→10, acquisition: trials 1→200; Session 11→20, reversal: trials 201→400). Bar indicates CS presentation (10 s), arrow bar shows reward delivery. Scattered plots show the $\Delta F/F$ signal across trials during the CS presentation. **d** Heatmaps illustrating averaged DA dynamics during CS presentation (bar) and reward delivery (arrow) across sessions. **e** (Left) Mean DA signal during CS+ (red) and CS- (blue) trials (bar) in acquisition (S1, S10) and reversal (S11, S20) sessions (arrow, reward delivery). (Right) Mean DA signal during CS presentation (one-way RM-ANOVA, $F(7,56) = 10.98$, $p < 0.0001$). **f** (Left) Mean DA signal during CS+ (red) and CS- (blue) trials (bar) during non-deterministic (S1, S10) and deterministic (S11, S20) sessions (arrow, reward delivery). (Right) Mean DA signal during CS presentation (one-way RM-ANOVA, $F(7,64) = 3.085$, $p = 0.0046$). **g** (Left) Relative increase of DA($\Delta$) during CS presentation in acquisition and reversal (one-way RM-ANOVA, Acq: $F(7,63) = 2.749$, $p = 0.0148$; Rev: $F(7,63) = 5.157$, $p = 0.0001$), and non-deterministic and deterministic sessions (one-way RM-ANOVA, Non-Det: $p > 0.05$, Det: $F(2.679,21.43) = 7.187$, $p = 0.0021$). Two-way RM-ANOVA SessionXContingency S1→S10: $F(9,135) = 3.459$, $p = 0.0007$; S11→S20: $p > 0.05$. (Right) Area under the curve (AUC) of DA during reward delivery (one-way RM-ANOVA, Acq: $F(3.170,22.19) = 11.61$, $p < 0.0001$; Rev: $F(2.886,20.20) = 7.778$, $p = 0.0013$; Non-Det: $p > 0.05$; Det: $F(2.780,22.24) = 7.665$, $p = 0.0013$). Two-way RM-ANOVA SessionXContingency, S1→S10: $F(9,135)=3.189$, $p = 0.0016$; S11→S20: $p > 0.05$. **h** Correlation analysis between the DA signal ($\Delta$) during CS presentation and the time ($\Delta$) mice approached the CS. (**i**) Correlation analysis between the DA signal (AUC) during reward collection and time mice approached the CS+. At least otherwise indicated, a total of $N = 8$ mice ($n = 4\male$, $n = 4\female$) were used for acquisition and reversal training sessions, and $N = 9$ mice ($n = 5\male$, $n = 4\female$) used for non-deterministic and deterministic sessions. Post-hoc Tukey's test: ***$p < 0.0001$, **$p < 0.001$, *$p < 0.05$, ns: non-significance. No adjustments were made for multiple comparison analyses. Data are presented as the mean ± SEM. Source data are provided as a Source Data file.

demonstrates that ACh, but not glutamate released from CINs, or ACh release from brainstem cholinergic neurons, plays an important role in encoding the cue-motivated incentive salience underlying approach behaviours.

## Conditional VAChTcKO mice display abnormal dopamine dynamics correlated with approach behaviours

Given our findings that mice with disrupted ACh release from CINs (VAChTcKOs) are unable to produce the approach behaviours present in their littermate counterparts (Fig. 3c and Supp. Fig. 3), and that DA signalling in the NAc underlies these behaviours (Fig. 2 and Supp. Fig. 2), we next tested whether DA signalling associated with cue-motivated approaches is affected in VAChTcKO mice. Previous experiments suggest that in general DA release should be decreased[49], but the use of photometry and GRAB sensors allows for evaluation of how ACh contributes to millisecond updating of DA signals that underlie behaviour. We used $\text{GRAB}_{DA}$[67] to record in vivo NAc DA dynamics in VAChTcKO ($N = 7$, $n = 3\male$, $n = 4\female$) and control littermate ($N = 8$, $n = 4\male$, $n = 4\female$) mice performing the Autoshaping task (Fig. 5a, b). Consistent with earlier reports indicating that striatal CINs modulate presynaptic DA release[18–21,75], and that VAChTcKO have DA deficits[49], we found that DA response amplitude during CS+ trials was reduced in VAChTcKO mice when compared to controls (Fig. 5c). Importantly, differences in DA dynamics during lit CS+ and CS- were significantly larger at late acquisition and reversal sessions in controls but not in VAChTcKO mice (Fig. 5d, left panel and Supp. Fig. 7a). This was a result of the decreased DA signalling during CS+, but also to the inability of DA levels during CS- to decrease in VAChTcKO mice across sessions (Fig. 5b, bottom-right heatmap). This abnormal DA signalling in VAChTcKO mice likely decreases signal to noise and contributes to the observed behavioural deficits. Similarly, the amplitude of DA responses during reward collection was blunted in VAChTcKO mice and did not change across sessions (Fig. 5d, middle panel and Supp. Fig. 7b).

The relative increase of DA response during CS presentation was correlated with approaches in both controls and VAChTcKO mice (Fig. 5d, right panel, and Fig. 5e), but with stronger correlation in controls (acquisition: $R^2 = 0.92$, reversal: $R^2 = 0.83$) compared to VAChTcKO mice (acquisition: $R^2 = 0.56$, reversal: $R^2 = 0.58$). Also, the DA response to reward was significantly correlated with approaches to the CS+ during presentation across all training sessions, but narrowly missed significance ($p = 0.0540$) in acquisition sessions in VAChTcKO mice

(Fig. 5f). Together, these findings suggest that despite VAChTcKOs exhibiting reduced NAc DA signalling during CS+ presentation and reward collection, they retain some ability to encode stimulus-reward associations, but to a lesser extent than control littermates (Fig. 5e, f), matching the late improvement in learning we observed (Fig. 3c and Fig. 5d, right panel). The observation that the amplitude of the DA signalling during CS- remains constant in VAChTcKOs across sessions when compared to controls (Fig. 5b) may also contribute to decrease the ability of mutant mice to discriminate between CS contingencies. These observations highlight the critical and likely constant updating of DA-ACh signals underlying reward-mediated behaviours, and indicate that cholinergic dysfunction leads to more subtle changes than merely decreasing DA, as previously suggested[49].

## Dysfunctional cholinergic signalling in the striatum drives abnormal direct and indirect spiny projecting neuron calcium dynamics

Previous reports have indicated that DA and ACh within the striatum often work in concert to regulate the activity and synaptic plasticity of SPNs from the direct and indirect pathways[20,23,24,26,42,82]. Indeed, evidence suggests concurrent dynamics in both SPN pathways regulate movement initiation, action selection, and/or behavioural reinforcement[3,83–86]. Importantly, it is suggested that altered DA-ACh balance may interfere with the coordinated activity of both SPN pathways and contributes to various neuropathologies including addiction and Parkinson disease[3,84,86]. VAChTcKO mice exhibit deficits in both ACh (Fig. 4) and DA dynamics (Fig. 5), underlying the close relationship between these two neurotransmitters. Thus, to understand the association between NAc ACh, DA, SPNs, and behaviour, we simultaneously studied the calcium activity of putative D1- and D2-SPNs during the acquisition of cue-motivated approach behaviours in the Autoshaping task, in both VAChTcKO and control mice.

To simultaneously characterise the influence of the direct (D1-SPN) and indirect (D2-SPN) pathways, we used multicolour photometry to monitor the fluorescence of green (GCaMP6s) and red (jRCaMP1a) genetically-encoded calcium indicators predominantly expressed in D1-SPNs and D2-SPNs of the same mice (Fig. 6a, b, Supp. Fig. 8)[87–89]. Targeting the expression of jRCaMP1a in D2-SPNs was achieved using a Cre-On adeno-associated virus (AAV) injected into D2-Cre[90] control mice ($N = 9$, $n = 5\male$, $n = 4\female$) or D2-Cre,VAChT$^{flox/flox}$ mice (VAChTcKO, $N = 10$, $n = 5\male$, $n = 5\female$). Simultaneously, expression of GCaMP6s in

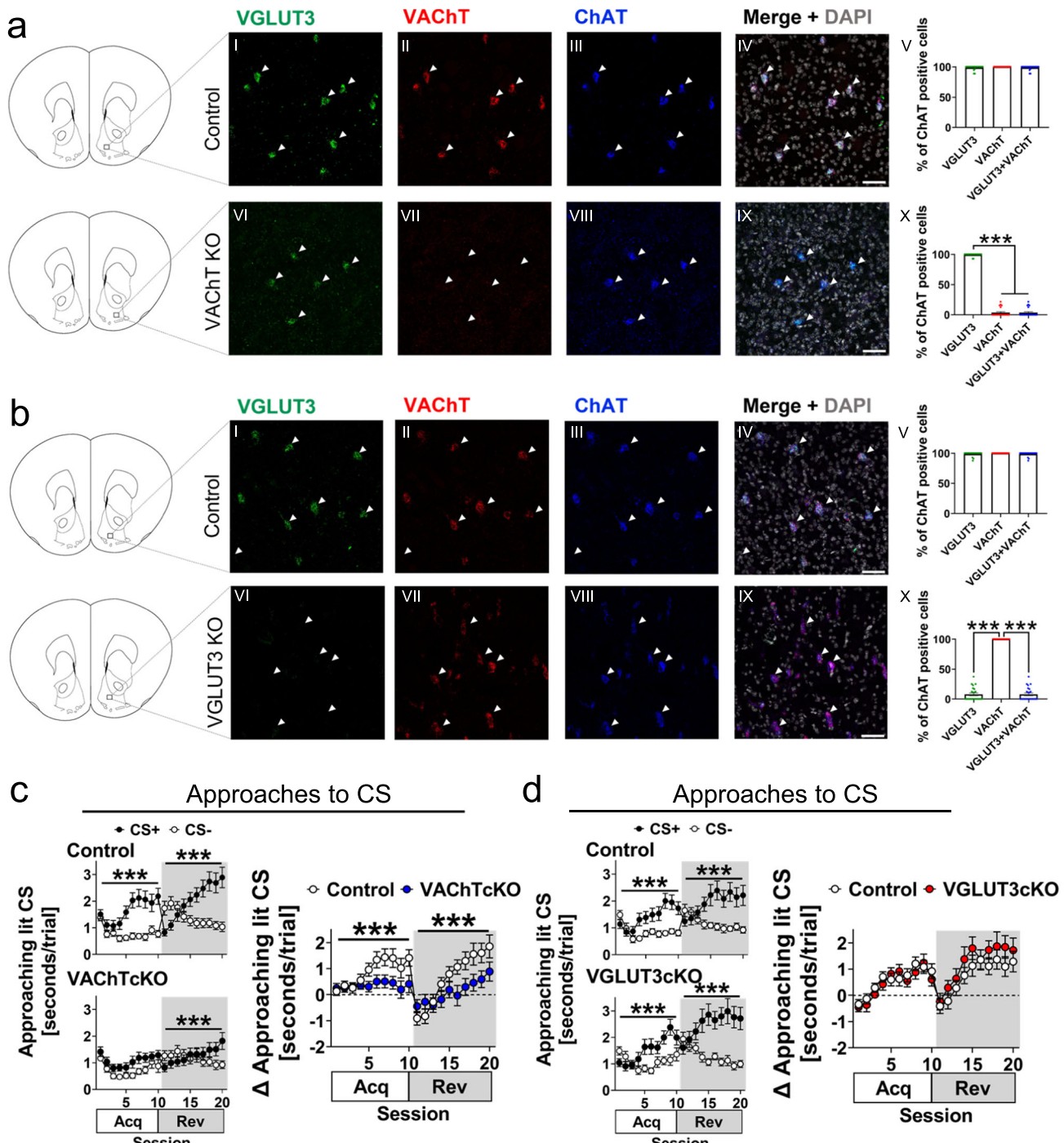

D1-SPNs was achieved by co-injection of a Cre-Off AAV within the same mice. Given that 95% of striatal cells are SPNs, and previous reports using this approach have demonstrated that fluorescence arising from interneurons is minimal[87,91,92], we assigned signals generated by jRCaMP1a to the indirect D2-SPN pathway and by GCaMP6s to the direct D1-SPN pathway. Finally, considering recent observations by Legaria et al.[93] indicating that calcium dynamics recorded from striatal SPNs may not reflect spiking-related events but instead may be non-somatic (dendritic) changes, we interpreted our calcium recordings as likely arising from dendritic neuronal sub-structures.

Similar to wild-type C57BL/6j (Fig. 2a and Supp. Fig. 1a), VAChT[flox/flox] (Fig. 3c, Supp. Fig. 3a, Supp. Fig. 5d and Supp. Fig. 6a) and VGLUT3[flox/flox] (Fig. 3d and Supp. Fig. 4a), control (D2-Cre) mice performing the Autoshaping task spent more time approaching the CS+ than the CS- across acquisition and reversal sessions (Fig. 6c), whereas VAChTcKO mice showed impaired approach behaviours toward the stimuli, reproducing data in (Fig. 3c and Fig. 5d). The calcium activity of putative D1-SPNs was characterised by multi-phasic events during CS+ presentation and reward delivery (Fig. 6d), but a monophasic event in D2-SPNs during reward delivery across training sessions (Fig. 6e). These D1- and D2-SPN calcium events were severely disrupted in VAChTcKOs (Fig. 6d–f). Regarding D1-SPNs, we first observed during the CS+ and CS- onset a phasic calcium increase across all training sessions in both control and VAChTcKO mice (Fig. 6g). In VAChTcKOs, this event was significantly larger during the first two acquisition sessions in the CS+ compared to the CS-. Second, the calcium signal amplitude significantly reduced as approaches toward the CS+ increased in controls but not in VAChTcKOs (Fig. 6h, top

**Fig. 3 | Acetylcholine release from cholinergic interneurons, but not glutamate, is required to regulate approach behaviours toward reward-predicting cues. a** (Top) Triple-fluorescence in situ hybridisation (RNAscope) revealed that mRNA transcripts for the vesicular glutamate transporter type 3 (*slc17a8*, VGLUT3-green), the vesicular acetylcholine transporter (*slc18a3*, VAChT-red), and the choline acetyltransferase transporter (*chat*, ChAT-blue) simultaneously express in interneurons from the nucleus accumbens of control littermate mice ($N = 3$ VAChT$^{flox/flox}$ mice, $n = 24$ cells/mouse). The boxed area from the depicted coronal brain section is enlarged in the right panels (I–IV) showing cholinergic interneurons abundantly expressing VGLUT3, VAChT and ChAT (arrowheads); nuclei were stained with DAPI (grey). (V) All ChAT+ cholinergic interneurons co-expressed transcripts for VGLUT3 and VAChT ($p > 0.05$). (bottom) In contrast, VAChTcKO mice ($N = 3$ mice, $n = 24$ cells/mouse; panels VI–IX) did not express transcripts for VAChT (VII), despite (X) VGLUT3 and ChAT mRNA were visualised in cholinergic interneurons (arrowheads) (one-way ANOVA, $F(2,66) = 2065$, $p < 0.0001$). Scale bar–50μm. **b** (Top) Nucleus accumbens neurons from control littermate mice showed abundant expression of VGLUT3 (green), VAChT (red) and ChAT (blue) transcripts ($N = 3$ VGLUT3$^{flox/flox}$ mice, $n = 24$ cells; panels I–IV). Nuclei were labelled with DAPI (grey). (V) All ChAT+ cholinergic interneurons also expressed VGLUT3 and VAChT mRNA. (bottom) No VGLUT3 transcript (arrowheads) was observed in neurons expressing VAChT and ChAT transcript in VGLUT3cKO mice ($N = 3$, $n = 24$ cells/mouse; panels VI–X). One-way ANOVA, $F(2,69) = 830.8$, $p < 0.0001$. Scale

bar–50μm. **c** In contrast to control littermate mice ($N = 24$ VAChT$^{flox/flox}$ mice, $n = 13♂$, $n = 11♀$), VAChTcKO mice ($N = 25$, $n = 11♂$, $n = 14♀$) performing the Autoshaping task showed deficits to approach the CS+ during acquisition (Acq) and reversal (Rev) sessions (two-way RM-ANOVA SessionXCS interaction, Acq-VAChTcKO: $p > 0.05$; Rev-VAChTcKO: $F(9,432) = 7.197$, $p < 0.0001$; Acq-Control: $F(9,414) = 8.299$, $p < 0.0001$; Rev-Control: $F(9,414) = 17.96$, $p < 0.0001$). Moreover, the relative time (Δ) mice approached reward-predicting CS revealed that VAChTcKO (blue circles) were impaired when compared to control littermate mice (blank circles). Two-way RM-ANOVA SessionXGenotype interaction, Acq: $F(9,423) = 3.203$, $p = 0.0009$; Rev: $F(9,423) = 4.232$, $p < 0.0001$. **d** VGLUT3cKO mice ($N = 23$, $n = 12♂$, $n = 11♀$) spent more time approaching the CS+ across acquisition and reversal sessions, similar as their control littermate mice ($N = 24$ VGLUT3$^{flox/flox}$, $n = 12♂$, $n = 12♀$). Mixed-effects model SessionXCS interaction, Acq-VGLUT3cKO: $F(9,394) = 7.212$, $p < 0.0001$; Rev-VGLUT3cKO: $F(9,395) = 9.091$, $p < 0.0001$; Acq-Control: $F(9,414) = 6.897$, $p < 0.0001$; Rev-Control: $F(9,414) = 10.68$, $p < 0.0001$. The relative time (Δ) mice approached reward-predicting CS showed that VGLUT3cKO (red circles) were similar than control littermate mice (blank circles) across sessions (two-way RM-ANOVA SessionXGenotype interaction, Acq: $p > 0.05$; Rev: $p > 0.05$). Post-hoc Tukey's test: ***$p < 0.0001$, **$p < 0.001$, *$p < 0.05$. No adjustments were made for multiple comparison analyses. Data are presented as the mean ± SEM. Source data are provided as a Source Data file.

panels). Finally, following reward delivery the calcium signal in control mice was characterised by a bi-phasic burst (Fig. 6i) and pause event (Fig. 6j, top-left panel) across acquisition and reversal sessions. The amplitude of the phasic (burst) calcium increase was larger in control than VAChTcKO mice. Interestingly, we found the amplitude of the pause mechanism after reward delivery significantly increased as mice spent more time approaching the CS+ (Figs. 6c and 6j). In contrast, despite VAChTcKO mice also showing a bi-phasic burst-pause response in D1-SPNs after delivery of rewards, the amplitude of the pause mechanisms was significantly reduced when compared to control mice. Together, our findings suggest that the calcium activity of putative D1-SPNs during the CS+ presentation and reward delivery progressively decreased as reward predictability increased during the Autoshaping task. Moreover, the activity of D2-SPNs in control mice significantly increased after reward delivery (Fig. 6j, bottom-left panel). Surprisingly, we found this reward-evoked activity instead decreased in VAChTcKO mice. Also, although no phasic D2-SPN activity was observed after CS- offset in either control or VAChTcKO mice (Figs. 6e and 6j, bottom-right panel), the calcium signalling amplitude was reduced in VAChTcKOs when compared to control mice across all training sessions. Our findings suggest that an adequate balance of DA-ACh within the NAc is critical for regulation of the coordinated calcium activity of the direct and indirect SPN pathways underlying cue-motivated approach behaviours[28,94].

### Acetylcholine released from nucleus accumbens cholinergic interneurons is necessary to regulate approach behaviours

Previous reports have highlighted that neurons from the NAc[54,58–61,66], but not the dorsal striatum[65,95], encode the acquisition of Pavlovian approach behaviours[54,58–61,66]. It has also been suggested, however, that lesions in the dorsal striatum of rats facilitate responses to the food reward magazine[60] and contribute to incentive salience[96].

Although NAc ACh may be required for approach behaviour, VAChTcKO mice have reduced ACh release in the dorsal striatum as well[49,76], which might contribute to the behavioural deficits we observed. Moreover, the behavioural disruption may reflect developmental adaptations of affected circuits due to genetic inactivation of VAChT early in development. To test these possibilities, we rescued the ability of CINs in adult VAChTcKOs to release ACh by local injection of an AAV-VAChT within the NAc ($N = 10$, $n = 5♂$, $n = 5♀$; Fig. 7a–d). Additionally, we co-injected AAV-ACh3.0 in the same group of mice to monitor ACh dynamics during the performance of the Autoshaping task. Alternatively, VAChTcKO mice co-injected with AAV-mCherry and

AAV-ACh3.0 ($N = 11$, $n = 5♂$, $n = 6♀$) were used as negative controls (sham). We found that relative to the sham control group, mice with rescued expression of VAChT in the NAc approached the CS+ more than the CS- (Fig. 7e), similar to control littermate mice ($p > 0.05$). Moreover, consistent with the notion that NAc CIN pauses are necessary for the processing of incentive salience and synaptic plasticity[9,11,13,27,73,97], VAChT-rescued mice showed significantly decreased ACh signalling across acquisition and reversal sessions after reward delivery (Fig.7f–h). This pause event was not observed in sham VAChTcKOs, consistent with our earlier experiments (Fig.4). Our findings suggest that the observed behavioural deficits in VAChTcKO mice are due to altered local circuitry mechanisms in the NAc specifically. These deficits can be rescued during adulthood by restoring the potential of CINs to generate a brief ACh salience-evoked pause response, most likely by maintaining 'optimal' cholinergic tone within the region.

## Discussion

Using a combination of automated touchscreen testing, fibre photometry, genetically-encoded sensors, and genetic mouse lines with deletion of VAChT or VGLUT3 transporters in striatal CINs, we revealed the dynamics of dopaminergic-cholinergic signalling underlying cue-motivated approach behaviours. Specifically, we report that a constant interaction and updating between ACh and DA signalling are critical to coordinate circuit mechanisms regulating the calcium activity of the direct and indirect SPNs underlying approaches to reward-predicting cues. Moreover, we demonstrate that interfering with ACh release from CINs alters the balance of DA-ACh dynamics and disrupts the activity of both SPN pathways, leading to a profound impairment in learning associations between CS+ and rewards, reflected as impaired development of approach behaviours toward CS locations during the Autoshaping task. Notably, we also show that restoring the expression of VAChT in CINs selectively in the NAc rescued ACh dynamics and approach behaviours. These results provide direct evidence that ACh released from NAc CINs, but not glutamate, plays a prominent role in the transference of incentive salience from rewards toward environmental stimuli predicting rewards.

Substantial literature suggests that CINs act as "gatekeepers" of striatal circuitry, exerting a considerable influence over cortical excitatory inputs, as well as modulating synaptic plasticity mechanisms in striatal outputs to regulate behaviour[45]. CINs are tonically active and electrophysiological evidence has demonstrated that they temporarily pause their firing in response to salient sensory cues associated with

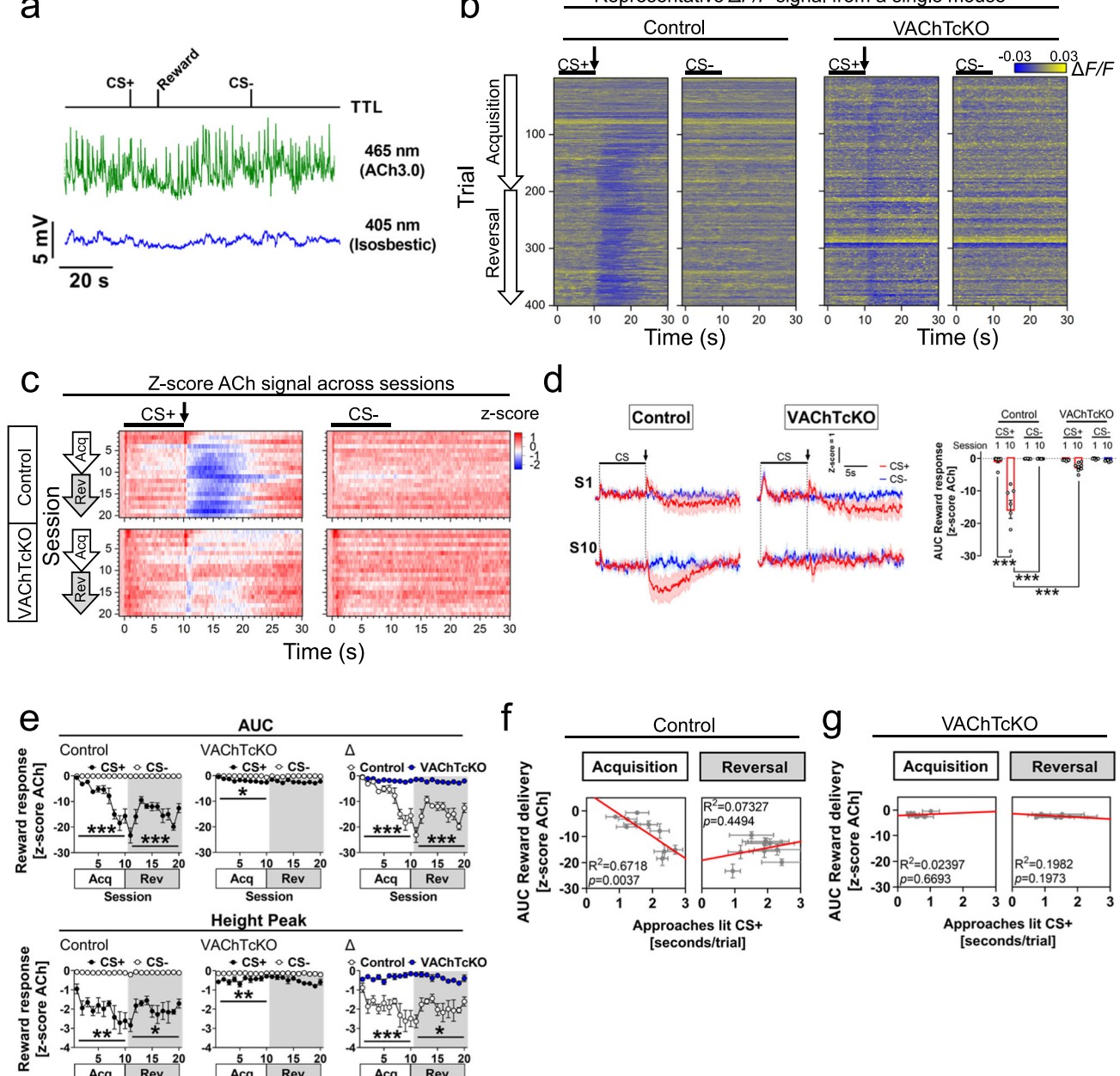

**Fig. 4 | Deficits in acetylcholine release from cholinergic interneurons underlie impaired approach behaviours to reward-predicting stimuli. a** Representative fibre photometry recording from a control littermate mouse showing raw fluorescence signal at 465nm (green, ACh3.0) and isosbestic 405nm (blue). A TTL input signal was used to time-lock CS onset and reward delivery. **b** Representative trial-by-trial heatmaps showing nucleus accumbens ACh dynamics ($\Delta F/F$) from a control littermate and VAChTcKO mouse performing the Autoshaping task (Session 1→10, acquisition: trials 1→200; Session 11→20, reversal: trials 201→400). Bar indicates CS presentation (10 s), arrow bar shows reward delivery. **c** Heatmaps illustrating trial average ACh dynamics (z-score) across acquisition (Acq, S1→S10) and reversal (Rev, S11→S20) sessions (CS+, left panels; CS-, right panels). Bar indicates CS presentation (10 s) and arrow bar reward delivery. **d** (left) Trial average of CS+ (red) and CS- (blue) ACh recordings during acquisition sessions (S1, S10) in control littermate and VAChTcKO mice. Bar indicates CS presentation (10s), arrow bar indicates reward delivery. (right) Area under the curve (AUC) of ACh signal (z-score) after CS+ offset (reward delivery) and CS- offset. In contrast to the first training session (S1), The ACh signal significantly reduced in trained control mice (S10), but VAChTcKO (one-way ANOVA, F(7,52) = 30.31, $p < 0.0001$. **e** (Top panels) AUC of ACh signal after CS

offset (CS+, reward delivery - filled circles; CS-, no reward - blank circles) across sessions in control (top-left panel, two-way RM-ANOVA SessionXCS, Acq: F(9,108) = 18.43, $p < 0.0001$; Rev: F(9,108) = 8.338, $p < 0.0001$), VAChTcKO (top-middle panel, Acq: F(9,126) = 2.527, $p = 0.0107$; Rev: $p > 0.05$), and relative ($\Delta$) AUC differences between genotypes (top-right panel, Acq: F(9,117) = 16.61, $p < 0.0001$; Rev: F(9,117) = 8.878, $p < 0.0001$). (bottom panels) Height peak of ACh signal in control mice (bottom-left, two-way RM-ANOVA SessionXCS, Acq: F(9,108) = 4.091, $p = 0.0002$; Rev: F(9,108) = 2.491, $p = 0.0126$) and VAChTcKO mice (bottom-middle, Acq: F(9,126) = 3.426, $p = 0.0008$; Rev: $p > 0.05$). The relative ($\Delta$) amplitude of events were larger in control than VAChTcKO mice across sessions (bottom-right, Acq: F(9,117) = 5.438, $p < 0.0001$; Rev: F(9,117) = 2.450, $p = 0.0136$). **f** Correlation analysis between the ACh AUC signal during reward delivery and time control mice spent approaching lit CS+. **g** Correlation analysis between ACh response during reward delivery and time VAChTcKO mice approached lit CS+. At least otherwise indicated, a total of N = 7 (n = 4♂, n = 3♀) control littermate mice and N = 8 (n = 4♂, n = 4♀) VAChTcKO mice were used. Post-hoc Tukey's test: ***$p < 0.0001$, **$p < 0.001$, *$p < 0.05$. No adjustments were made for multiple comparison analyses. Data are presented as the mean ± SEM.

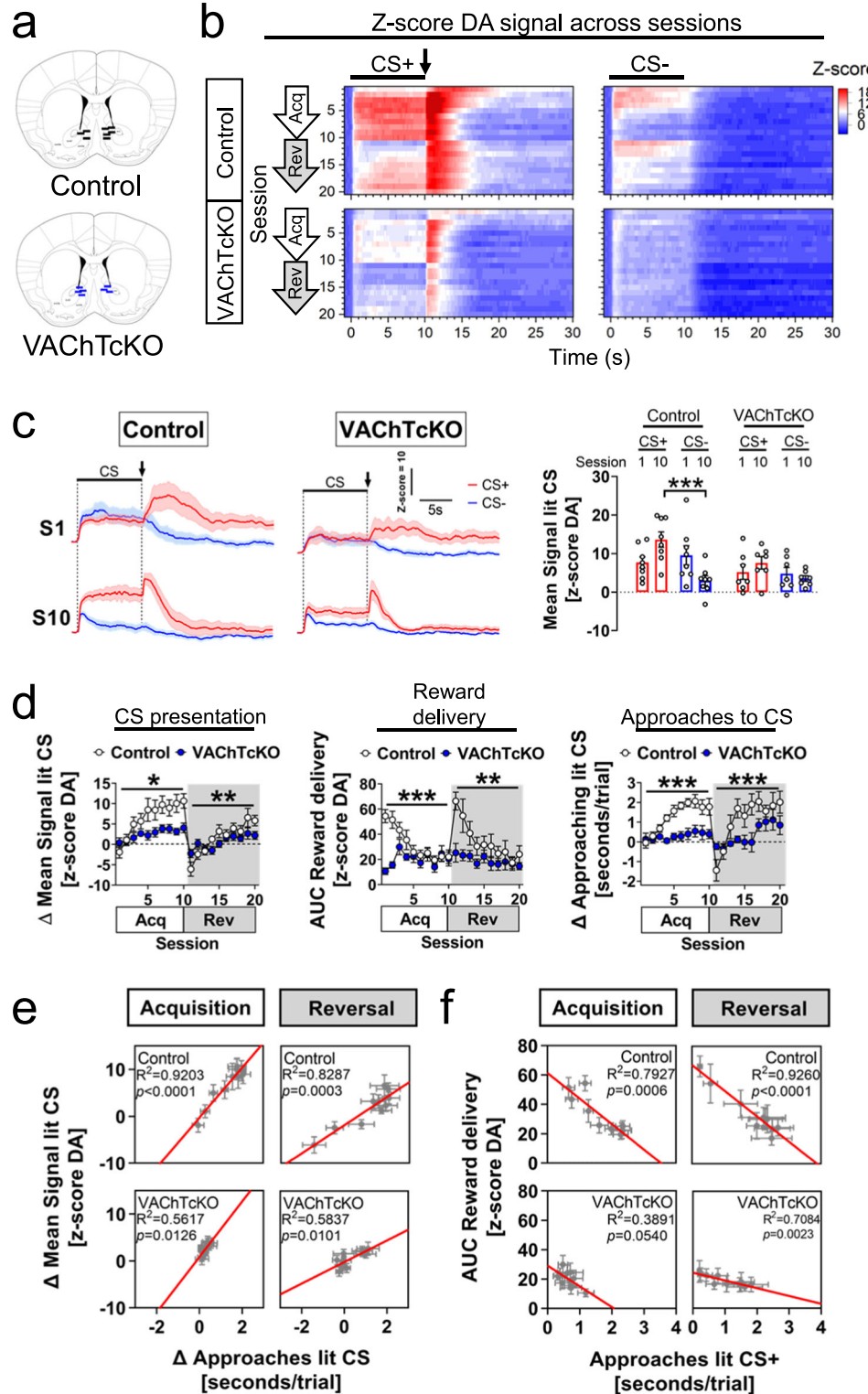

rewards[8,9,40]. This pause mechanism, which should transiently decrease cholinergic tone, is thought to gate striatal input information, encode synaptic plasticity mechanisms, and modulate striatal output to the basal ganglia[45] by reducing nicotinic receptor modulation of DA release[20,21,98], as well as modulating the activity of D1- and D2-SPNs via nicotinic and muscarinic receptors[14,22,42,99]. Moreover, striatal GABAergic interneurons appear to play a fundamental role in the modulation of the network SPN activity mediated by CIN-dependent disynaptic inhibitory mechanisms[26,27]. Although the neural mechanisms regulating the pause in firing are not completely understood, it

has been shown to depend on the activation of D2 receptors expressed in CINs[13,100–102], and activation of excitatory inputs from intralaminar thalamic nuclei[103,104].

Despite electrophysiological evidence suggesting the role of CIN pauses in the regulation of striatal circuitry function and behaviour, it has only been with the recent development of genetically-encoded biosensors[105] that in vivo recordings of extracellular ACh with sub-second resolution have been achieved[77]. This is particularly important considering that tonically-active CINs also co-release glutamate, and in vivo electrophysiological methods cannot distinguish the

**Fig. 5 | Mice with deficits in acetylcholine release show abnormal dopamine dynamics in nucleus accumbens. a** Schematic brain sections depicting location of fibre stub tips implanted within the nucleus accumbens of control littermate (top-black bar) and VAChTcKO (bottom-blue bar) mice. **b** Heatmaps illustrating trial average DA signal (z-score) from acquisition (Acq, S1→S10) and reversal (Rev, S11→S20) sessions (CS+, left panels; CS-, right panels). Bar indicates the CS presentation (10 s), arrow bar indicates reward delivery. **c** (Left panel) Averaged DA signal (z-score) from CS+ (red) and CS- (blue) trials during acquisition sessions (S1, S10) in control littermate and VAChTcKO mice. Bar indicates CS presentation (10 s) and arrow bar the reward delivery. (right panel) Mean DA signal during CS+ (red bars) and CS- (blue bars) presentation at S1 and S10 acquisition sessions in control and VAChTcKO mice (one-way ANOVA, $F_{(7,52)} = 4.292$, $p = 0.0008$). Scattered data points represent individual mice. **d** (Left panel) Mean DA signal amplitude (Δ) during CS presentation between control littermate and VAChTcKO mice (two-way RM-ANOVA SessionXGenotype interaction, Acq: $F_{(9,117)}=2.400$, $p=0.0156$; Rev:

$F_{(9,117)} = 2.699$, $p = 0.0069$). (middle panel) Area under the curve (AUC) of DA signal during reward delivery in control littermate and VAChTcKO mice (two-way RM-ANOVA SessionXGenotype interaction, Acq: $F_{(9,117)} = 6.758$, $p < 0.0001$; Rev: $F_{(9,117)} = 2.843$, $p = 0.0046$). (right panel) Relative time (Δ) mice approached the CS during presentation. In contrast to controls (blank circles), VAChTcKO mice (blue circles) did not discriminate between CS stimuli across sessions (two-way RM-ANOVA SessionXGenotype interaction, Acq: $F_{(9,117)} = 4.399$, $p < 0.0001$; Rev: $F_{(9,117)} = 3.888$, $p = 0.0002$). **e** Correlation analysis between the mean DA signal (Δ) during CS presentation, and the time mice spent approaching the CS stimuli (Δ). **f** Correlation analysis between the DA response during reward collection (AUC) and time mice spent approaching the CS+. A total of $N = 8$ ($n = 4♂$, $n = 3♀$) control littermate mice and $N = 7$ ($n = 3♂$, $n = 4♀$) VAChTcKO mice were used. Post-hoc Tukey's test: \*\*\*$p < 0.0001$, \*\*$p < 0.001$, \*$p < 0.05$. No adjustments were made for multiple comparison analyses. Data are presented as the mean ± SEM.

consequences of ACh and glutamate release. Consistent with a previous report using a Go/No-Go task in mice[102], we observed that NAc ACh dynamics in control mice performing the Autoshaping task are mainly characterised by a transient reward-evoked decrease in cholinergic tone, likely reflecting the pause of activity of CINs electrophysiologically detected in vivo. This profound decrease of cholinergic signals correlates with the acquisition of approach behaviours, suggesting a relevant gating mechanism necessary for the acquisition or maintenance of cue-motivated learning behaviours. Supporting this hypothesis, we report that mice with disrupted ACh release (VAChTcKO) from CINs are profoundly impaired in learning to approach CS predicting rewards. Mice with disrupted glutamate release (VGLUT3cKO) from CINs perform similarly to controls. Reduced VAChT expression decreases ACh release to levels below the detection limit of microdialysis[49]. Because striatal baseline ACh levels are very low in VAChTcKO, CIN pauses in these mutants likely lack the potential to neuromodulate local plasticity mechanisms within the NAc. Thus, our fibre photometry observations in control mice strongly suggest that the reward-evoked ACh dynamics correlate with findings using in vivo electrophysiological approaches[8,9,40]. Conversely, we suggest that the anomalous endophenotype in VAChTcKO mice results from disturbed ACh storage from cholinergic synapses leading to a blunted vesicular ACh release from CINs[76,106,107], that reduces ACh signal-to-noise ratio. Together, these data strongly suggest that ACh released from CINs plays critical roles in the neuromodulation of striatal circuits underlying cue-motivated learning behaviours.

The potential of individuals to transfer the reinforcing and motivational properties of rewards toward environmental stimuli seems to depend on DA within the NAc[108]. For example, mesolimbic DA depletion with 6-hydroxydopamine[66] or DA receptor antagonism within the NAc[109] impairs both acquisition and performance of appetitive Pavlovian approach behaviour. Consistent with this idea, we demonstrated that approach behaviours during the Autoshaping task robustly correlate with rapid increases in NAc DA signalling across acquisition and reversal training sessions, but inversely correlate during reward collection. Furthermore, our findings suggest DA signalling in NAc encodes the level of certainty with which mice can predict rewards. Thus, during the Autoshaping non-deterministic contingencies in which the probability of receiving rewards from each CS is 50%, DA signalling does not correlate with approach behaviours during CS+ presentation or reward delivery. The observation that DA signalling in VAChTcKO mice can still weakly correlate with approach behaviours when compared to controls suggest deficits in transferring the motivational incentive salience of rewards toward approach behaviours, yet mice are still able, at least to a certain degree, to associate the presentation of CS+ with rewards (however, this phenotype does not manifest as approach behaviours towards the location of CS). Supporting this hypothesis, previous reports have shown that striatal VAChTcKO mice are able to learn complex contingencies leading to

rewards when performing training-intensive touchscreen-based behavioural tasks such as the heterogeneous sequence task, the pairwise visual discrimination task, and the 5-choice serial reaction time task (5-CSRTT)[49,75]. Finally, we observed a ~1s long DA response at the onset of the CS presentation, even to the CS- (Fig. 2e, f). Several lines of evidence suggest that in addition to the reward prediction error, DA responses are also observed during arousing sensory and/or novel events[110–113]. However perhaps more likely is that this is a conditioned response that briefly generalises. Both CS+ and CS- are similar stimuli (large bright rectangles) that differ only in their spatial location. It is perhaps not surprising that following conditioning, when a large bright stimulus appears, there is a generalised DA response even to the CS-, which is rapidly curtailed once the system identifies the stimulus as the CS-. This explanation is similar to the idea that two sensory systems pass information to reward circuitry: a "low road", which provides rapid but low-resolution information, and a "high road" that provides high resolution information that becomes available following a brief delay[114–116].

Within the striatum, the integration and output of information to the rest of the basal ganglia relies on the activity of GABAergic SPNs, which constitute as much as 95% of the entire neuronal population within the region[3]. SPNs are divided into two equally-sized and molecularly distinct subpopulations segregated by their output projection pathways through the basal ganglia. SPNs of the direct pathway express $G_{s/olf}$-coupled D1 DA receptors (D1-SPNs) whereas SPNs of the indirect pathway express $G_{i/o}$-coupled D2 dopamine receptors (D2-SPNs)[94,117,118]. Additionally, both SPN subpopulations express cholinergic $G_q$-coupled M1 and $G_i$-coupled M4 receptors, with M4 being more abundant on D1-SPNs[28]. Although still controversial[3,83,86,99], a recent model proposes that the activity of the direct and indirect SPN pathways 'compete' to determine the animal's behavioural response, via modulating synaptic plasticity at inputs onto SPNs[83]. In this work, by recording calcium dynamics simultaneously from both SPN subpopulations[87,92], we observed that in control mice, the calcium activity of D1-SPNs was characterised by biphasic events during CS presentation and reward delivery, while D2-SPN activity manifested as a single reward-evoked event. Previous reports[119,120] using a combination of electrophysiological recordings and optogenetic manipulation partially agree with our findings indicating that during Pavlovian conditioning tasks, D1-SPNs from dorsomedial striatum increase their activity as a function of reward value, while activity of D2-SPNs is reduced. However here we report, to our knowledge for the first time, that the rapid reward-evoked increase in D1-SPN calcium activity is followed by a pause response that opposes an increase of calcium in D2-SPNs, suggesting that during the Autoshaping task, the concurrent calcium dynamics of the direct and indirect SPN pathways are mutually necessary to encode the acquisition and maintenance of approach behaviours[60,121]. Consistent with this idea, we observed that in contrast to control mice, VAChTcKOs showed abnormal calcium activity from

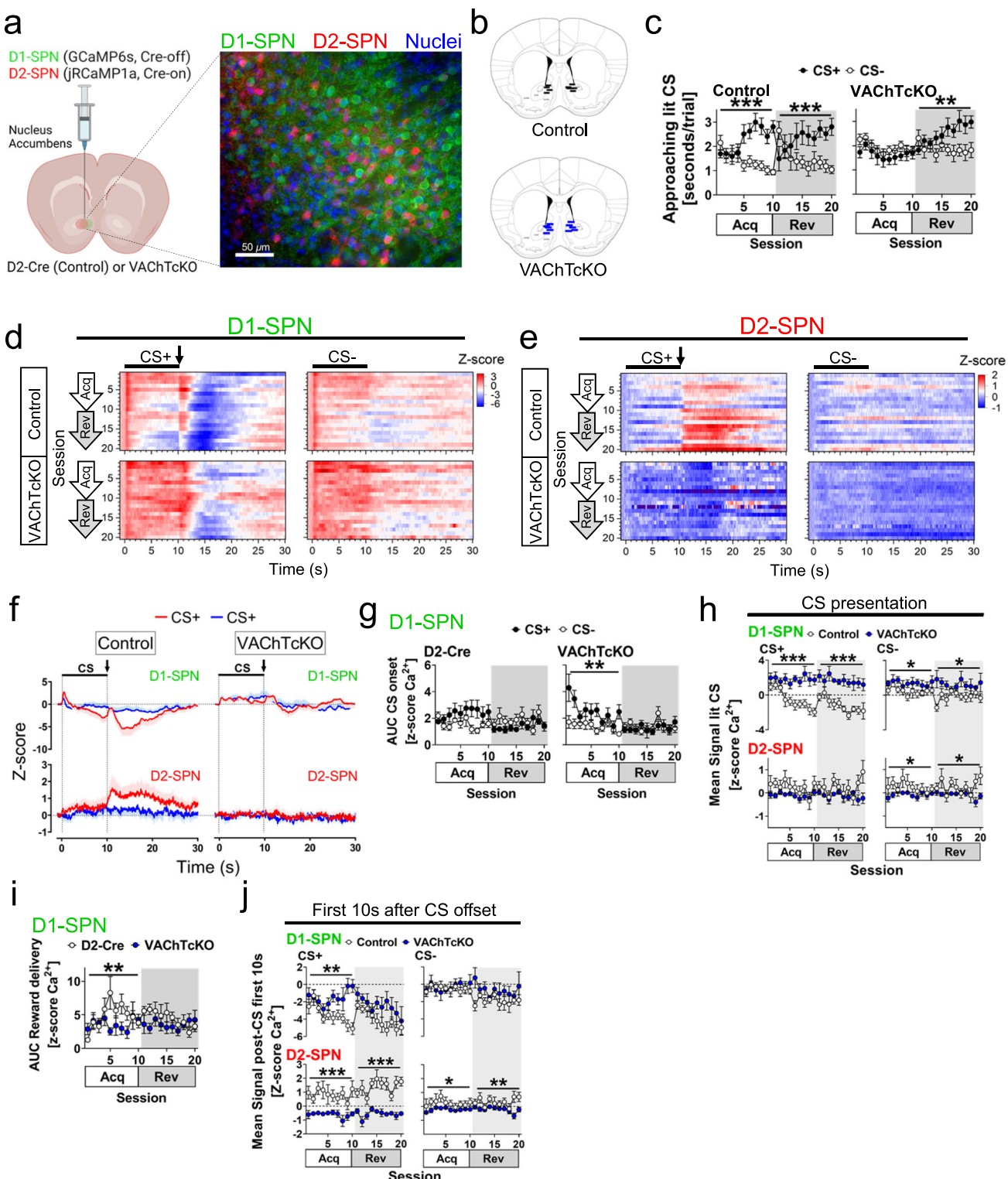

the direct and indirect SPN pathways, that likely underlies the observed deficits in learning the associations between CS and rewards. Previous reports have suggested that the activity of SPNs during stimuli conferring incentive salience heavily relies on the co-occurrence of rapid increases in DA release and cessations of ACh release[3,122], but also on the differential expression of dopaminergic and cholinergic receptors among D1-SPNs and D2-SPNs[28,94]. For example, recent work demonstrated that the inhibition of D1-SPNs mediated by M4 receptors is indirectly regulated by the modulation of D2 receptors expressed in CINs[23]. It is plausible that the observed calcium

hyperactivity in D1-SPNs of VAChTcKOs may be due to a reduced signalling of M4 receptors expressed within this subpopulation of neurons. In contrast, hypoactivity of D2-SPNs may be the result of a reduced activation of M1 receptors expressed in D2-SPNs. It is important to highlight that a recent elegant work from Legaria et al.[93] demonstrated that the striatal SPN calcium fibre photometry signal may reflect not only spiking dynamics, and instead much of the signal may arise from the dense dendritic arborisation of neurons. Therefore, following the suggestions of Legaria et al. (2022), we interpret the observed SPN calcium signals, regulated by continuous updating of

**Fig. 6 | Acetylcholine release from cholinergic interneurons regulates the concurrent calcium activity of spiny projecting neurons. a** (Left) Schematic representation of control (D2-Cre) and VAChTcKO mice receiving an unilateral nucleus accumbens injection of an AAV mix (1:1) of GCaMP6s,Cre-Off and jRCaM-P1a,Cre-On constructs to respectively express GCaMP6s in D1-SPNs, and jRCaMP1a in D2-SPNs. (right) Representative nucleus accumbens immunoreactivity staining showing expression of GCaMP6s in D1-SPNs (green), and jRCaMP1a in D2-SPNs (red). Nuclei stained with Hoechst (blue). Scale bar−50μm. **b** Schematic brain sections showing location of fibre track lesions within the nucleus accumbens in control (top, black) and VAChTcKO mice (bottom, blue). **c** Control mice spent more time approaching the CS+ than the CS- during acquisition (Acq, S1→S10) and reversal sessions (Rev, S11→S20) (two-way RM-ANOVA SessionXCS interaction, Acq: $F_{(9144)} = 8.640$, $p < 0.0001$; Rev: $F_{(9144)} = 5.883$, $p < 0.0001$). In contrast, VAChTcKO mice spent similar time approaching the CS+ and CS- during acquisition sessions, but visited more the CS+ than CS- during reversal sessions (two-way RM-ANOVA SessionXCS interaction, Acq: $p > 0.05$; Rev: $F_{(9,162)} = 3.515$, $p = 0.0005$). **d** Heatmaps showing trial average calcium dynamics (z-score) from D1-SPNs across sessions. Bar indicates CS presentation (10 s) and arrow bar reward delivery. **e** Heatmaps showing trial average calcium dynamics (z-score) from D2-SPNs across sessions. **f** Trial average calcium dynamics (z-score) during CS+ (red) and CS- (blue) from D1-SPNs (top) and D2-SPNs (bottom) at the last acquisition session (S10). **g** Area under the curve (AUC) calcium signal in D1-SPN during CS+ (filled circles) and CS- (blank circles) onset. No differences between CS stimuli were observed in control mice ($p > 0.05$). However, a larger CS+ onset response was observed during the first acquisition sessions in VAChTcKOs (two-way RM-ANOVA CS factor, Acq: $F_{(1,18)} = 9.153$, $p = 0.0073$; Rev: $p > 0.05$). **h** Mean calcium signalling (z-score) from

D1-SPNs (top panels) and D2-SPNs (bottom panels) during the CS stimuli presentation (10 s). The calcium D1-SPN dynamics significantly reduced across sessions during the CS+ in control mice (blank circles). In VAChTcKOs (blue circles), the signal amplitude remained elevated across sessions (two-way RM-ANOVA genotype, Acq: $F_{(1,17)} = 19.88$, $p = 0.0003$; Rev: $F_{(1,17)} = 20.12$, $p = 0.0003$). Additionally, the calcium amplitude was elevated during CS- trials in VAChTcKOs when compared to control mice (two-way RM-ANOVA genotype, Acq: $F_{(1,17)} = 5.140$, $p = 0.0367$; Rev: $F_{(1,17)} = 4.855$, $p = 0.0416$). The calcium signal in D2-SPNs was significantly reduced during CS- trials in VAChTcKO mice (two-way RM-ANOVA genotype, Acq: $F_{(1,17)} = 4.933$, $p = 0.0402$; Rev: $F_{(1,17)} = 5.580$, $p = 0.0304$). **i** AUC calcium response during reward delivery. A larger calcium response was observed in control mice when compared to VAChTcKOs (two-way RM-ANOVA CSXGenotype interaction, Acq: $F_{(9,153)} = 3.304$, $p = 0.0010$; Rev: $p > 0.05$). **j** Mean calcium dynamics (z-score) from D1-SPN (top panels) and D2-SPNs (bottom panels) during the first 10 s after the CS offset. After CS+ offset, the calcium dynamics reduced more in D1-SPNs from controls (blank circles) than VAChTcKOs (blue circles) during acquisition sessions (two-way RM-ANOVA genotype, Acq: $F_{(1,17)} = 5.486$, $p = 0.0059$; Rev: $p > 0.05$). The calcium dynamics in D2-SPNs were reduced in VAChTcKOs during CS stimuli (two-way RM-ANOVA genotype, CS+ Acq: $F_{(1,17)} = 30.38$, $p < 0.0001$; CS+ Rev: $F_{(1,17)} = 70.43$, $p < 0.0001$; CS- Acq: $F_{(1,17)} = 6.702$, $p = 0.0191$; CS- Rev: $F_{(1,17)} = 11.30$, $p = 0.0037$). A total of $N = 9$ ($n = 5♂$, $n = 4♀$) D2-Cre control and $N = 10$ ($n = 5♂$, $n = 5♀$) VAChTcKO mice were used. Post-hoc Tukey's test: $***p < 0.0001$, $**p < 0.001$, $*p < 0.05$. No adjustments were made for multiple comparison analyses. Data are presented as the mean ± SEM. Figure 6a, left panel, was created with BioRender.com.

---

ACh and DA, as possibly reflecting a dendritic 'eligibility trace'. The eligibility trace is posited by emerging theories of synaptic plasticity as a kind of flag, set at the synapse by the co-activation of pre- and postsynaptic neurons, that leads to weight change in a susceptible synapse only if an additional factor such as novelty, punishment, or reward is present[123]. Moreover, this additional factor is often implemented by the phasic activity of neuromodulators such as DA and ACh[124–126]. Therefore, our observations may reflect a mechanism of eligibility trace for synaptic plasticity and behavioural conditioning[124,125,127], mediated by a continuous updating of ACh and DA dynamics, and triggered by behaviourally relevant stimuli. Future work is needed to address how the heterogeneous contribution of dopaminergic and muscarinic receptors expressed in D1- and D2-SPNs may regulate the shape, volume, and stability of dendritic spines[128], and how this could influence changes in synaptic plasticity mechanisms regulating behaviour.

The use of transgenic mice chronically affecting CIN function is a valuable tool for understanding relevant endophenotypes associated with brain disorders, and specifically in this case to separate the contributions of VAChT or VGLUT3-mediated neurotransmitter release for behaviour. An important issue is that developmental compensatory mechanisms often hinder the interpretation of how acute factors affect the release of ACh and/or glutamate from CINs and how behaviours are related to effects arising from chronic manipulations. For example, the expression of VAChT in the cortex is reduced by 50% in D2-Cre mice[49], which could potentially contribute to the observed behavioural phenotypes in VAChTcKO mice. Moreover, although our findings support the idea that the NAc[54,58–61,66] circuitry is critical for the acquisition of Pavlovian approach behaviours (but[65,95]), others suggest that contributions from the dorsal striatum may facilitate incentive salience[96] and responses to collect rewards[55,62]. Because the re-expression of VAChT within the NAc of adult VAChTcKO mice restored reward-evoked decreased cholinergic tone (i.e. ACh pauses) and approach behaviours comparable to control littermate mice during the Autoshaping task, it seems likely that the maturation of striatal network activity mechanisms underlying cue-motivated approach behaviours does not require ACh released from CINs[129]. Furthermore, our evidence strongly indicates that ACh, but not glutamate, released from CINs within the NAc, is necessary for the regulation of approach

learning behaviours. Finally, although our study does not directly demonstrate that inserting VAChT into the NAc of VAChTcKO mice increases extracellular ACh baseline levels, it is important to highlight that in the striatum, only cholinergic interneurons require VAChT to transfer ACh from the cytoplasm into synaptic vesicles[130]. Currently, there is no other function for VAChT that we are aware of. Taken together, our work suggests that an intricate balance between DA-ACh, and its reciprocal rapid updating during learning, is critical for the regulation of local network mechanisms and neuronal engrams underlying approaches to reward-predicting cues. Our observations shed new light on DA-ACh balance[122] which has been proposed as an aetiological mechanism underlying a variety of brain-related disorders including addiction, anxiety, obsessive-compulsive disorders, schizophrenia and Parkinson's disease.

## Methods
### Animals
Adult wild-type C57BL/6j (8 weeks old) were directly obtained from The Jackson Laboratories (strain#000664, Bar Harbor, ME). Mutant VAChT^flox/flox and VGLUT3^flox/flox mouse lines[131,132] were backcrossed to C57BL/6j background for at least 8 generations and maintained as inbreed strains in our mouse colony. LoxP sequences flanking the VAChT gene do not interfere with cholinergic marker expression, and VAChT^flox/flox and VGLUT3^flox/flox mice do not differ behaviourally from wild-type C57BL/6j mice[131,133]. D2-cre mice [Tg(Drd2-cre) 44Gsat; GENSAT obtained from MMRRC B6/129/ Swiss/FVB mixed background (strain#32108)][90] and Engrailed-1 (En1)-Cre mice [Jackson stock#En1tm2(cre)Wrst/J, strain#007916; 129S1/SvImJ mixed background][134] were backcrossed for at least four generations to C57BL/6j upon arrival to our laboratory. VAChT^flox/flox and VGLUT3^flox/flox mice were crossed with D2-Cre mice to generate D2-Cre;VAChT^flox/flox (hereafter VAChTcKO) and D2-Cre;VGLUT3^flox/flox (VGLUT3cKO), respectively. Striatal cholinergic interneurons simultaneously co-express VAChT, VGLUT3 and D2 receptors[74]. We used genetically modified mice in which striatal VAChT or VGLUT3 was selectively eliminated from cholinergic interneurons using the Cre-Lox recombination approach as previously described[49]. We elected to use D2-Cre so we could use the same Cre line to delete VAChT and VGLUT3. One alternative would be to use a different Cre

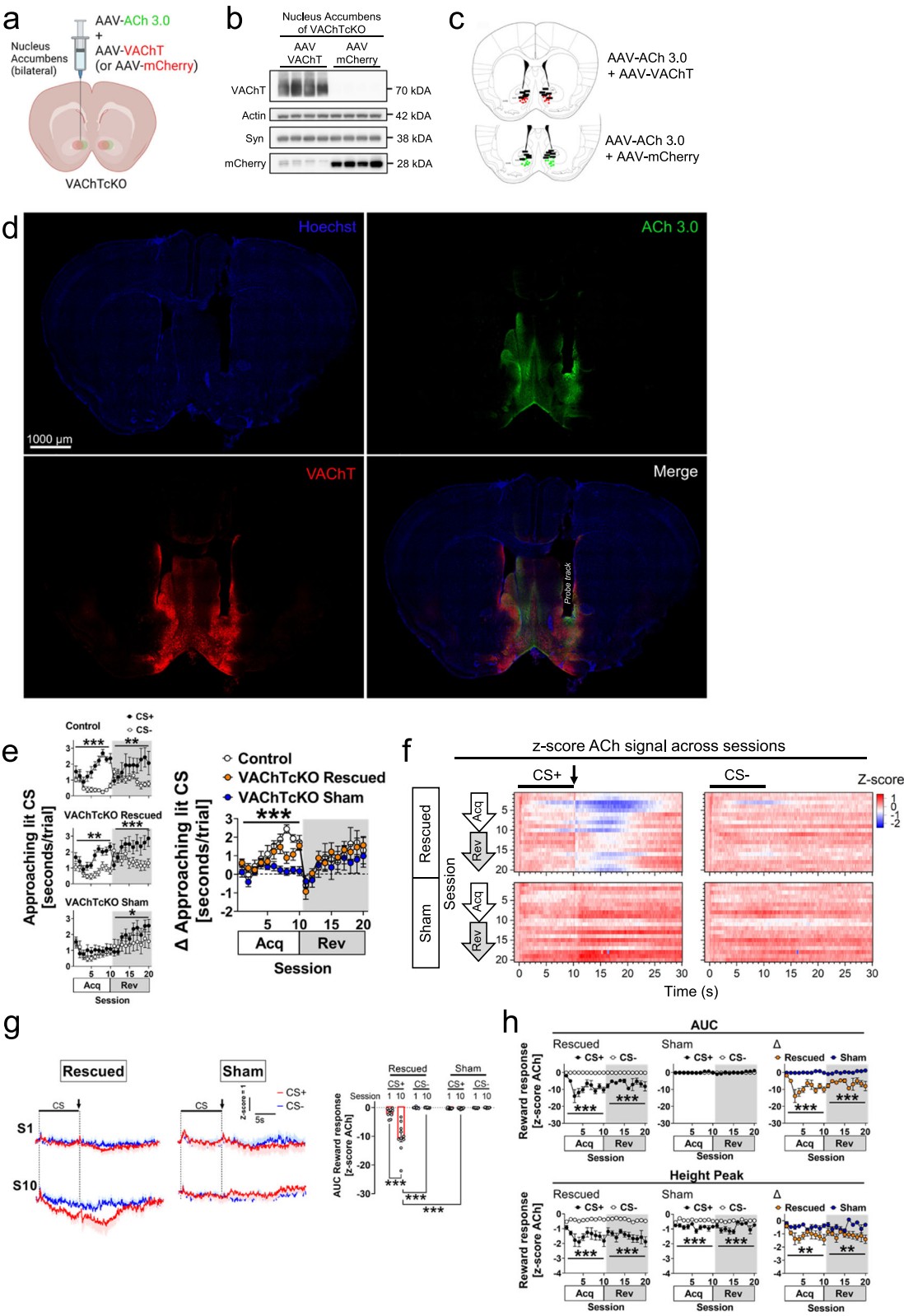

driver mouse line (i.e. ChAT-Cre and VGlut3-Cre). However, these lines would not be selective, and they would generate full KOs of VAChT or VGLUT3. Moreover, we are aware of ectopic expression of Cre in any line used[135], hence the D2-Cre line offers the best compromise to use the same line to delete both genes. D2-Cre mice behaved essentially as control mice in the Autoshaping task. Separate cohort of VAChT[flox/flox] mice were crossed with En1-Cre mice to generate En1-Cre,VAChT[flox/flox] mice. Cohorts of mice used in the

present study were generated by breeding littermates VAChTcKO and VGLUT3cKO to VAChT[flox/flox] and VGLUT3[flox/flox], respectively. Additionally, breeding littermates En1-Cre,VAChT[flox/flox] and VAChT[flox/flox] were used. The use and care of the animals was conducted in agreement with the Canadian Council of Animal Care guidelines and the animal protocols approved by the Animal Care and Veterinary Services (ACVS) from Western University (protocols #2020-162, 2020-163).

**Fig. 7 | Deficits in approach behaviours in VAChTcKO mice are rescued after expressing VAChT within the nucleus accumbens. a** VAChTcKO (Rescued) mice received bilateral nucleus accumbens injections of 1:1 AAV-VAChT.mCherry (red) and AAV-ACh3.0 (green). This approach allowed to rescue the expression of VAChT within the nucleus accumbens of VAChTcKO mice and simultaneously record ACh dynamics during the Autoshaping task using fibre photometry. Alternatively, an independent group of VAChTcKO (Sham) mice received injections of a 1:1 AAV-mCherry and AAV-ACh3.0. **b** Western blot analysis demonstrated that VAChTcKO mice receiving AAV-VAChT.mCherry injections ($N = 4$ mice), but not AAV-mCherry injections ($N = 4$ mice), showed immunoreactivity for VAChT protein expression within the nucleus accumbens. Immunoreactivity for mCherry was found in both treatments. Expression of actin and synaptophysin (Syn) were used as a protein loading control. **c** Schematic brain sections from VAChTcKO mice depicting AAV injection site of ACh3.0+VAChT (red circles), ACh3.0+mCherry (green circles) and tip of fibre optic tracks (black bars). **d** Representative immunostaining from a VAChTcKO-rescued mouse receiving bilateral injections of AAV-ACh3.0 + AAV-VAChT.mCherry. Immunoreactivity for ACh3.0 (green) and VAChT (red) was observed within the nucleus accumbens in both hemispheres. Nuclei were stained with Hoechst (blue). The tip of the fibre optic track was also located within the nucleus accumbens. GFP and mCherry immunoreactivity was reproduced in all mice tested in this study (see below). for Scale bar–1000μm. **e** (Left panels) VAChTcKO-rescued but VAChTcKO-Sham mice spent more time approaching the CS+ across sessions (Rescued, two-way RM-ANOVA SessionXCS interaction, Acq: F(9,162)=3.073, $p = 0.0020$; Rev: F(9,162) = 5.304, $p < 0.0001$. Sham, Acq: $p > 0.05$; Rev: F(9,180) = 2.224, $p = 0.0225$, similar as control littermate mice (Control, two-way RM-ANOVA SessionXCS interaction, Acq: F(9,108) = 6.398, $p < 0.0001$; Rev: F(9,108) = 3.027, $p = 0.0029$). Control mice shown here correspond to experiment of Supp. Fig. 5d. (right panel) Similar as controls (blank circles, $p > 0.05$),

VAChTcKO-rescued mice (orange circles) spent more time (Δ) approaching the reward-predicting CS+, while the VAChTcKO-Sham mice (blue circles) did not discriminate both CS stimuli (two-way RM-ANOVA SessionXTreatment interaction, Acq: F(18,225) = 2.749, $p = 0.0003$; Rev: $p > 0.05$). Compared to controls, VAChTcKO-Sham mice were impaired during acquisition sessions (two-way RM-ANOVA SessionXTreatment interaction, Acq: F(9,144) = 4.646, $p < 0.0001$; Rev: $p > 0.05$). **f** Heatmaps illustrating trial average ACh dynamics ($z$-score) across sessions. Bar indicates CS presentation and arrow bar reward delivery. **g** (Left) Trial average ACh signal ($z$-score) during the CS+ (red) and CS- (blue) at early (S1) and late (S10) acquisition sessions in VAChTcKO-Rescued and VAChTcKO-Sham mice. Bar indicates CS presentation (10 s). Arrow bar indicates reward delivery. (Right) Area under the curve (AUC) after CS stimuli offset from individual mice (circles) (one-way ANOVA, F(7,76) = 36.28, $p < 0.0001$). **h** (Top) AUC of ACh dynamics after CS+ offset (filled circles) and CS- trials (blank circles) across sessions in VAChTcKO-Rescued (two-way RM-ANOVA CS factor, Acq: F(1,18)=51.78, $p < 0.0001$; Rev: F(1,18) = 46.77, $p < 0.0001$), VAChTcKO-Sham (Acq: $p > 0.05$; Rev: $p > 0.05$), and relative (Δ) difference between treatments (Rescued-orange circles; Sham-blue circles) (Acq: F(1,19) = 57.22, $p < 0.0001$; Rev: F(1,19) = 52.60, $p < 0.0001$). (bottom) Similarly, ACh height peaks showed differences between contingencies in VAChTcKO-Rescued (two-way RM-ANOVA CS factor, Acq: F(1,18) = 33.12, $p < 0.0001$; Rev: F(1,18) = 28.09, $p < 0.0001$), VAChTcKO-Sham (Acq: F(1,20) = 35.92, $p < 0.0001$; Rev: F(1,20) = 19.46, $p = 0.0003$), and relative (Δ) differences between treatments (Acq: F(1,19) = 11.12, $p = 0.0035$; Rev: F(1,19) = 9.304, $p = 0.0066$). A total of $N = 10$ VAChTcKO-rescued ($n = 5$♂, $n = 5$♀) and $N = 11$ VAChTcKO-Sham ($n = 5$♂, $n = 6$♀) mice were used. Post-hoc Tukey's test: ***$p < 0.0001$, **$p < 0.001$, *$p < 0.05$. No adjustments were made for multiple comparison analyses. Data are presented as the mean ± SEM. Figure 7a was created with BioRender.com.

## Experimental design

Experiments were performed on 3- to 7-month-old male and female mice. Unless otherwise indicated, animals were housed in groups of two to four per cage at 22–23 °C, 50 ± 10% humidity, with a 12:12h reverse light-dark cycle. Food and water were provided ad libitum until behavioural testing, at which point mice were mildly food restricted (90–95% of their original body weight) to increase their motivation to perform a behavioural task. Experiments were performed during the dark cycle (between 9:00 a.m. and 6:00 p.m.).

To characterise how the co-transmission of acetylcholine and glutamate from striatal CINs, or cholinergic pedunculopontine/laterodorsal tegmental nuclei neurons projecting to the striatum impacts on Pavlovian approach associative learning behaviours, independent cohorts of wild-type C57BL/6j mice, or mutant mice and their corresponding control littermates were used to perform an automated touchscreen Autoshaping task (Fig. 1). Separate cohorts of wild-type and mutant mice were used for fibre photometry experiments and behavioural testing. Mice with head implants were single housed to prevent infection of surrounding incision area or damage to the implant.

## Touchscreen Autoshaping task

Experiments were conducted using automated Bussey–Saksida Mouse Touchscreen Systems Model 80614-20 (Lafayette Instruments, Lafayette, IN). The touchscreen Autoshaping task has been previously described[51,52]. Experiments were carried out inside sound-attenuating cabinets consisting each in a standard operant chamber and a touchscreen 12.1-inch monitor. The operant chamber was trapezoidal-shaped constructed from three black Plexiglas walls, which open to the touchscreen (Dimensions: 20 cm x 18 cm screen-reward tray x 24 cm at screen) (Fig. 1a). The ceiling of the chamber was made of clear Plexiglas and the floor of perforated stainless steel with a waste tray situated below. The chamber was equipped with a liquid reward dispensing magazine located centrally in front of the touchscreen and linked to a liquid reward dispenser pump (strawberry milkshake, Neilson Dairy). A light emitting diode illuminated the food magazine during reward delivery. Computer graphic white square stimuli were presented on

the touchscreen at either left or right side of the reward magazine. A miniature infrared camera was installed above the chamber to allow monitoring of the animals' behaviour. Animal activity was recorded via infrared photobeams located in front of each side of the screen (approaches), entries to the reward magazine (reward collection latency) and opposite side of the screen (trial initiation) (Fig. 1a). Schedule design, control of the apparatus via Whisker control system, and data collection used ABET II Video Touch software V21.02.26.

The Autoshaping task consisted of two pre-training phases followed by ten consecutive acquisition sessions and ten reversal sessions (one session per day, seven days a week). The first pre-training phase consisted of a unique session in which the animal habituated to the operant chamber by remaining inside of it for 30 min. No action was triggered regardless of mouse's behavioural status. The second pre-training phase consisted of at least two consecutive sessions (30 min-long each) in which reward (~7 μl) was delivered after a variable ITI (0–30 s; additional time allowed if necessary to ensure animal is not in magazine when ITI ends), with the magazine illuminated and a tone (1 s, 3 kHz, 80 dB) emitted upon delivery. The animal must enter the magazine to collect the reward (upon which the magazine light extinguished) to initiate the next delay period. Criteria was reached when the animal collected at least 50 rewards during the session. Animals not able to reach criteria after four consecutive pre-training sessions were excluded from the study. In total, 3 VGLUT3cKO ($n = 2$♂, $n = 1$♀), 1♀ control VGLUT3, and 1♂ VAChTcKO were excluded.

On the day after pretraining, animals were trained to associate the presentation (10 s) of a conditioned stimuli (CS) with the delivery of 10 μl of strawberry milkshake reward pumped into the central magazine. During a trial, a stimulus on one side of the screen (e.g., right) was designated to anticipate the delivery of a reward (CS+), while the opposite side screen (e.g., left) did not lead to reward contingencies (CS−). The location of CS+ and CS− were counterbalanced across mice but once designated, they remained constant across consecutive trials at least otherwise indicated. A single CS contingency was presented per trial. After a variable inter-trial interval (ITI, 45–90 s), the mouse initiated a trial by breaking the back infrared beam (BIR) within the chamber (Fig. 1b). During CS onset, a click tone (0.2 s, 2 kHz, 80 dB)

was generated to maximise the probability that the animal will be able to see both sides of the screen upon stimulus presentation and minimise inadvertent stimulus approaches. Upon CS+ offset a tone (1 s, 3 kHz, 80 dB) was emitted, a reward delivered to the magazine, and a light inside the reward magazine illuminated until the first nose poke for reward collection was registered via a light infrared beam located inside the reward magazine. Upon CS− offset, no reward was delivered and no tone, or light inside the reward magazine was generated. Following CS offset (and, if reward was delivered, entry into the magazine for reward collection), a new variable ITI began. The house light remained off throughout the session. A full session consisted in 40 trials, including 20 presentations of each CS contingency delivered in a pseudorandom order that no more than two similar CS trials were repeated consecutively. Sessions ended by completing 40 trials or 60 min, whichever reached first. Mice were trained 1 session per day. In total, mice underwent 10 consecutive acquisition sessions, followed by 10 reversal sessions, in which the pre-determined location of CS+ and CS- trials for each animal was reversed (CS+ becomes CS−, and CS− becomes CS+). It is anticipated that by reversing the contingency of the task, animals must adapt their reward prediction behavioural performance accordingly.

To evaluate the Pavlovian nature of the task, an independent cohort of wild-type C57BL/6j mice underwent 10 non-deterministic acquisition sessions, in which each side of the screen had 50% probability to be either CS+ or CS− (Fig. 1c). A total of 40 trials (20 CS+ and 20 CS−) presentations within 60 min were delivered per session. Under this non-deterministic contingency, animals were unable to predict what stimulus (left or right screen) anticipated the delivery of a reward. After the completion of the non-deterministic acquisition sessions, animals were trained for another 10 sessions with deterministic contingencies as previously described in Fig. 1b.

The primary performance measure in this task is the time mice spent in front of the CS+ and CS− screens. However, visits to the reward magazine (latency time to collect rewards), and latency and number of touches to the CS+ and CS− screen were also recorded.

## Viral vectors

For experiments to record extracellular dopamine or acetylcholine using fibre photometry, expression of GRAB$_{DA2m}$[67] and GRAB$_{ACh3.0}$[77] was achieved via injection of either AAV9-hSyn-GRABDA$_{2m}$ ($3.1 \times 10^{13}$ gc/ml, Vigene Biosciences, Inc, Rockville MD) or AAV9-hSyn-GRAB$_{ACh3.0}$ ($3.65 \times 10^{13}$ gc/ml, Vigene Biosciences Inc) into the NAc, respectively. To simultaneously record intracellular Ca$^{2+}$ from striatal D1-SPN and D2-SPN[87], a 1:1 mixture of AAV9-CBA.DO(FAS)-GCaMP6s (Cre-Off) ($5.1 \times 10^{12}$ gc/ml, UNC Vector Core, Chapel Hill, NC) (AAV was a gift from Bernardo Sabatini, Addgene plasmid # 110135; http://n2t.net/addgene:110135; RRID:Addgene_110135) and AAV1-Syn.Flex.NES-jRCaMP1a.WPRE.SV40 (Cre-On) ($2.1 \times 10^{13}$ gc/ml, Addgene, Watertown, MA) (AAV was a gift from Douglas Kim & GENIE Project, Addgene viral prep # 100846-AAV1; http://n2t.net/addgene:100846; RRID:Addgene_100846)[89] was injected within the NAc of D2-cre (control) or VAChTcKO mice. For combined experiments rescuing the expression of VAChT and fibre photometry, a 1:1 injection mixture of a custom-made undiluted AAV9-eSyn.mCherry-2A-mSLC18A3-WPRE (Vector Biosystems Inc, Malvern, PA) and AAV9-hSyn-GRAB$_{ACh3.0}$ (Vigene Biosciences) was done into the NAc of VAChTcKO mice. Alternatively, a separate cohort of mice received a 1:1 injection mixture of AAV9-hSyn-mCherry ($3.6 \times 10^{13}$ gc/ml, Addgene) (AAV preparation was a gift from Karl Deisseroth, Addgene viral prep # 114472-AAV9; http://n2t.net/addgene:114472; RRID:Addgene_114472) and AAV9-hSyn-GRAB$_{ACh3.0}$ as a control (sham) group within the NAc of VAChTcKO mice.

## Western blot

Mice were sacrificed by cervical dislocation, brains were removed and dissected on ice to isolate the NAc, then flash-frozen on dry ice before

transferring to -80 °C for long-term storage. The day of the experiment, the tissue was weighed and homogenised on ice-cold RIPA buffer (50 mM Tris, 150 mM NaCl, 0.1% SDS, 0.5% sodium deoxycholate, 1% triton X-100, pH 8.0) with phosphatase inhibitors (1 mM NaF and 0.1 mM Na$_3$VO$_4$) and protease inhibitor cocktail (Calbiochem, catalogue# 539134-1SET, 1:100). Extracts were rocked at 4 °C for 20 min and centrifuged at 10,000 x $g$ for 20 min at 4 °C to isolate protein. Protein was quantified using BioRad DC Protein assay (BioRad, Catalogue# 5000112). 25 µg of protein were loaded onto 4–12% Bis-Tris Plus Gels (ThermoFisher) and protein was transferred onto PVDF membranes by BioRad Semi-Dry Trans-Blot Turbo System. The primary antibodies used for immunoblotting were: rabbit anti-VAChT (Synaptic System, catalogue# 139103, 1:2000), rabbit anti-synaptophysin (Cell Signaling Technology, catalogue# 5461S, 1:3000), rabbit anti-mCherry (Abcam, ab167453, 1:2000) and mouse anti-β-actin (Sigma-Aldrich, catalogue# A3854, 1:25000) as loading control. The secondary antibody was goat anti-rabbit HRP (BioRad, Catalogue# 170-6515, 1:7500). Proteins were visualised using chemiluminescence with the ChemiDoc MP Imaging System (BioRad).

## RNA in situ hybridisation (ISH)

Fluorescent multiplex ISH was performed using RNAScope (Advanced Cell Diagnostics, ACD, Newark, CA) to detect VGLUT3 (Slc17a8, probe Mm-Slc17a8-#431261), ChAT (CHAT, probe Mm-Chat-C2, #408731-C2), and VAChT (Slc18a3, probe Mm-Slc18a3-C3, #448771-C3) mRNA transcripts from fresh frozen 10 µm-thick brain sections of VAChTcKO, VGLUT3cKO and their corresponding control littermate mice. Hybridisation procedure followed previously described instructions from the manufacturer and elsewhere[136]. Briefly, brain sections were incubated with RNAscope protease digestion for 15 min at 40 °C, followed by hybridisation for 2h with a mixture containing target probes against mouse VGLUT3, VAChT and CHAT mRNAs. Then, hybridisation signals were detected using the following protocol: AMP1 Atto550 (30 min incubation), AMP2 Alexa488 (15 min), and AMP3 Atto647 (30 min) were incubated at 40 °C, followed by 15 min (40 °C) with AMP4B amplification solution. Sections were co-stained with DAPI, mounted in Epredia Immu-Mount, #9990402) and images captured using the Leica DM6B Thunder imager (Leica Microsystems Inc.).

## Immunohistochemistry

Mice were anaesthetised with ketamine (100 mg.kg⁻¹)-xylazine (20 mg.kg⁻¹) and then transcardially perfused with ice-cold phosphate-buffered saline (PBS) followed by 4% paraformaldehyde (PFA). Brains were kept overnight in 4% PFA and then transferred into a PBS-azide solution, and a vibratome was used to cut 40 µm sections. After slicing, free-floating sections were rinsed with PBS and incubated in Tris-buffered saline (TBS) containing 1.2% Triton X-100 for 20 min. The sections were rinsed with TBS and blocked for 1 h in TBS containing 5% (v/v) normal goat serum at room temperature. After blocking, sections were rinsed twice with TBS and then incubated overnight at 4 °C with chicken anti-GFP (Abcam, ab13970 1:500) and rabbit anti-mCherry (Abcam, ab167453, 1:200) in TBS containing 0.2% Triton X-100 and 2% normal goat serum. The following day after ~18 h incubation with the primary antibodies, sections were washed twice for 10 min each in TBS and then incubated for 1 h at room temperature with Alexa 488 goat anti-chicken (Thermo Fisher, A11039, 1:500) and Alexa 633 goat anti-rabbit (Thermo Fisher, A21070, 1:500) antibodies in TBS 0.2% Triton X-100 and 2% normal goat serum. The sections were washed twice in TBS for 10 min and then incubated with Hoechst 33342 (Thermo Fisher H3570, 1:1000) to counterstain the nuclei. Images were captured using the Leica DM6B Thunder imager (Leica Microsystems Inc.)

## Surgical procedures and fibre photometry

Viral infusions and optic fibre implants were carried out as previously described[75]. Briefly, mice were anaesthetised with 5% isoflurane

induction rate and placed in a stereotaxic frame, after which anaesthesia was maintained at 1.5–3%. A heating pad was placed under the mice to maintain body temperature (37 °C). The top of the skull was exposed, and holes were drilled for viral infusion needle, optic fibre implant, and two skull screws. Viral injections aiming the NAc were made using a microsyringe pump (0.5 μl, 0.1 μl/min) at the following coordinates from Bregma (AP: 1.8 mm, ML: 0.5 mm, DV: 4.0 mm)[137]. Injectors were left in place for 5 min and then slowly removed. Only mice receiving AAV9-eSyn.mCherry-2A-mSLC18A3-WPRE or AAV9-hSyn-mCherry were injected bilaterally, otherwise, counterbalanced unilateral viral injections were performed. Low-auto-fluorescence optic fibre implants (400 μm O.D, 0.48 NA, 5 mm-long, Neurophotometrics, San Diego, CA) were unilaterally inserted just above the injection site. Prior to experimentation, mice underwent a 3-weeks recovery period followed by food restriction (90-95% of their postrecovery body weight) for at least another two extra weeks.

Mice were first allowed to adapt to the touchscreen chamber and fibre patch-cord during the touchscreen pretraining sessions (see above). To record fluorescence signals, the photometry system was equipped with a fluorescent mini-cube (Doric Lenses, Quebec, Canada) to transmit sinusoidal 465 nm LED light modulated at 572 Hz and a 405 nm LED light modulated at 209 Hz. LED power was set at ~25 μW. Fluorescence was collected through the path-cord connected to the optic fibre implant of each mouse and transmitted back to the mini-cube, amplified, and focused into an integrated high sensitivity photoreceiver (Doric Lenses). Alternatively, for fibre photometry experiments requiring dual calcium recordings, a fluorescent mini-cube (Doric Lenses) able to transmit sinusoidal 465 nm LED (modulated at 572 Hz), 560 nm LED (modulated at 334 Hz), and a 405 nm LED (modulated at 209 Hz) was used. Fluorescent real-time signal was sampled at 12 kHz and then demodulated and decimated to 100 Hz using Doric Studio software V5.2.2.3 (Doric Lenses). The occurrence of behavioural manipulations including CS onset and reward delivery was recorded by the same system via TTL inputs from ABET II (Lafayette Instruments).

## Fibre photometry analysis

Analysis of the signal was done with a custom-written Phyton software available at Mousebytes (https://mousebytes.ca/comp-edit?repolinkguid=ccf27660-6442-4c90-a14c-cdbc663a7b72). Fluorescence signal from 405 nm, 465 nm, and 560 nm channels were low band-pass filtered to remove events exceeding 6 Hz. The isosbestic 405 nm channel was used to correct for bleaching and movement optical artifacts. Accordingly, any fluctuations occurring in the 405 nm isosbestic channel were removed from the 465 nm and 560 nm channels before analysis. For this purpose, the least-squares linear fit method[138] was applied to the isosbestic 405 nm signal to align it to the 465 nm signal (or 560 nm), producing a fitted 405 nm signal used to normalise the 465 nm as follows: $\triangle F/F = [465\ nm\ signal - (fitted\ 405\ nm\ signal)]/(fitted\ 405\ nm\ signal)$. Then, to assess changes from baseline fluorescence signal after CS onset within trials, the baseline z-score of the $\triangle F/F$ was calculated as follows: $z\ score = [\triangle F/F - (\mu baseline)]/\sigma baseline$, where $\mu\ baseline$ is the mean of $\triangle F/F$ values from baseline period (averaged signal collected 1s before CS onset) and $\sigma baseline$ is the standard deviation of $\triangle F/F$ values from baseline period.

Validation of in vivo acetylcholine dynamics using GRAB$_{ACh3.0}$ and fibre photometry (Supp. Fig. 5c) was achieved by recording in an open field arena, the $\triangle F/F$ fluorescence signalling during 10 min in VAChT$^{flox/flox}$ mice (n=6) before administering systemic donepezil (1 mg.kg$^{-1}$, i.p.), and signal recorded for another 60 min. To validate in vivo dopamine dynamics ($n = 8$) using GRABDA$_{2m}$ (Supp. Fig. 2c), or calcium signalling from D1-SPNs and D2-SPNs using GCaMP6s and jRCaMP1a ($n = 8$, Supp. Fig. 8d), respectively, baseline fluorescence signal was recorded during 5 min before administering systemic saline

injections (i.p.) followed by consecutive 1 h long recordings. Finally, mice received an injection of cocaine (10 mg/kg, i.p.) and fluorescence recorded for another 1 h.

## Statistics, data collection, and analysis

Behavioural data was extracted from ABET II Video Touch software V21.02.26 (Lafayette Instruments). The data supporting these findings can be visualised and are freely available at MouseBytes (https://mousebytes.ca/home). Microscopy images were processed using ImageJ 64-bit V1.8 (NIH). The generation of heatmaps, estimation of area under the curve (AUC), and height peak analysis of events were obtained using OriginPro 2021 V9.8.0.200 (OriginLab Corporation, Northampton, MA). Briefly, heatmaps illustrating $\triangle F/F$ or z-score from GRAB$_{DA}$, ACh3.0, or calcium signalling consisted of individual trials, or averaged trials (20 trials/session). Each began with 1 s long baseline before CS onset, followed by 10 s long CS+ or CS- presentation followed by the delivery of a single reward (7 μl) after CS+ offset. DA and calcium signal during CS presentation was calculated by averaging the signal during CS presentation. To find peaks, calculate height, and to integrate the AUC, the averaged trial signal from each session was used to find and calculate peaks (curve) within the ROI, obtaining the baseline by estimation of end points weighted (15%) within the window search. Alternatively, for the estimation of number of peaks during validation experiments recording calcium (Supp.Fig.8d), the 1st derivative signal within the 10 min window search (saline vs. cocaine) was applied followed by the Savitzky–Golay method to smooth the signal. To calculate the AUC and height peak of the DA signal after reward delivery, the window search was set for the following 5 s after CS+ offset. The AUC and height of the peak of the reward response during ACh recordings were estimated by setting the window search for the 10 s after CS+ offset. Alternatively, the mean signal post-reward delivery for the calcium recordings in D1-SPN and D2-SPN was calculated by averaging the signal during the 10 s immediately post CS offset. All data were imported to GraphPad Prism V9.3.1 for Windows 10 (GraphPad Software, San Diego, CA) for statistical analysis. Initially, a normality test (D'Agostino-Pearson) was performed to determine whether parametric or non-parametric tests were appropriate. No assumptions or corrections were made prior to data analysis. Differences between two groups were always examined using a two-tailed Student's t-test, where $p < 0.05$ was considered significant and $p > 0.05$ was considered non-significant. Comparisons between multiple groups were performed using analysis of variance (ANOVA; one-way and two-way with repeated measures), followed by Sidak's multiple comparisons test. Alternatively, when a dataset had missing values, we compared groups using a linear mixed-effects ANOVA model. A simple linear regression analysis was used to correlate DA and ACh signalling with approaches to the CS presentation. Estimation of sample size in Autoshaping task experiments was estimated using a partial eta squared ($\eta_p^2$) power analysis for repeated measures two-way ANOVA (power = 0.9, α = 0.05)[139]. For experiments combining fibre photometry and touchscreens, we used standard sample sizes ($N = 7$–11) as previously reported[87,93]. All cell counting, ISH, and immunohistochemistry experiments were performed by an experimenter blind to the experimental condition. Estimation of sample size was estimated based on previous reports. All data were expressed as mean ± s.e.m. and $p < 0.05$ were considered as statistically significant.

## Reporting summary

Further information on research design is available in the Nature Portfolio Reporting Summary linked to this article.

## Data availability

The datasets generated and analysed in this study can be visualised and are freely accessible in the Mousebytes repository, https://mousebytes.ca/home, and complementary Mousebytes repository,

https://mousebytes.ca/comp-edit?repolinkguid=ccf27660-6442-4c90-a14c-cdbc663a7b72. Additionally, generated data from all figures are provided in their corresponding Source Data files. Source data are provided with this paper.

## Code availability

The phyton-based code for fibre photometry analysis can be freely accessed from the complementary Mousebytes repository, https://mousebytes.ca/comp-edit?repolinkguid=ccf27660-6442-4c90-a14c-cdbc663a7b72.

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

## Acknowledgements

M.S. received support funding from BrainsCAN (Canada First Research Excellence Fund), Accelerator Awards as well as support for the Rodent Cognition Research and Innovation Core. M.A.M.P., V.F.P., L.M.S., T.J.B., received support from the Canadian Institutes of Health Research (CIHR, PJT 162431, PJT 159781), Natural Science and Engineering Research Council of Canada (402524-2013 RGPIN; 03592-2021 RGPIN), Tanenbaum Open Science Institute (TOSI) and a BrainsCAN Canada First Research Excellence Fund Accelerator Awards as well as support for the Rodent Cognition Research and Innovation Core. M.A.M.P. is a Tier I Canada Research Chair in Neurochemistry of Dementia. L.M.S. is a Tier I Canada Research Chair in Translational Cognitive Neuroscience and a CIFAR Fellow in the Brain, Mind and Consciousness program. T.J.B. is a Western Research Chair. We thank Salah El Mestikawy for kindly sharing VGLUT3$^{flox/flox}$ mice, and Alexxai Kravitz and Andrew Holmes for helpful comments on the manuscript.

## Author contributions

M.S., Y.L., J.R., L.M.S., V.F. P., M.A.M.P., and T.J.B. designed experiments. M.S., O.P.-L., L.G.-C., A.M.C., and A.R. performed experiments. M.S., O.P.-L., A.M.C., G.K.K., and S.H.T. analysed data. S.M. developed a pyton-based code to extract and analyse fibre photometry data. D.P. designed behavioural schedules on ABET II® for touchscreens. M.S., M.A.M.P., and T.J.B. wrote the manuscript. All authors were involved in revising the manuscript for intellectual content. All authors read and approved the final version of the manuscript.

## Competing interests

T.J.B. and L.M.S. have established a series of targeted cognitive tests for animals, administered via touchscreen within a custom environment known as the "Bussey-Saksida touchscreen chamber". Cambridge Enterprise, the technology transfer office of the University of Cambridge, supported commercialisation of the Bussey-Saksida chamber, culminating in a license to Campden Instruments. Any financial compensation received from commercialisation of the technology is fully invested in further touchscreen development and/or maintenance. M.S., O.PL., L.G.C., A.M.C., G.K.K., S.H.T., A.R., S.M., D.P., Y.L., J.L., V.F.P., and M.A.M.P. declare no competing interests.
