## [Peer Review File · Nature Communications]

Continuous Cholinergic-Dopaminergic Updating in the Nucleus Accumbens Underlies Approaches to Reward-Predicting CuesREVIEWER COMMENTS

Reviewer #1 (Remarks to the Author):

This interesting study demonstrates the close interaction between the striatal cholinergic interneurons and dopamine release in the mouse nucleus accumbens, during a autoshaping task performance of Pavlovian paradigm. Even the hypothesis of the DA-ACh relation is not new, it is very remarkable that this investigation dissects the action of acetylcholine and glutamate along the different phases of a Pavlovian task, both of them neurotransmitters that are coreleased by ChIs in the nucleus accumbens. To measure the DA or ACh release authors used genetically-encoded sensors and fibre photometry experiments together with transgenic mice. In addition, by means of calcium indicators, authors measured the activity of D1- and D2- SPNs during the task performance.

A major concern is related with the measure of the D1- and D2- SPNs activity in both VACHTcKO and control mice. In a recent article, published as preprint, it was compared the activity of striatal SPNs recorded by calcium imaging using GCaMP6s (the same as the one used in this article) with respect to the action potentials recorded extracellularly by single unit in the same neurons (Legaria et al., 2021). Basically, they found that calcium activity does not correlate with the somatic action potentials on SPNs. Instead, calcium activity correlates with changes on fluorescence in the neuropil, reflecting the inputs that neurons receive, instead than the outputs (the action potentials). Thus, authors must demonstrate that the fluorescence changes recorded in SPN are directly reflecting action potentials. If finally, the calcium signal is related to the SPN inputs instead to their action potentials, authors must reinterpret their results.

Another important concern is related with number of animals that authors used to demonstrate that ACh but not glutamate is necessary for the acquisition of the task. This conclusion is based on the independent comparisons between VACHTcKO vs control mice and the VGLUT3cKO vs control mice. To that end, authors used 25 VACHTcKO and 19 VGUT3cKO, reaching the statistical significance in the first group. Samples are unbalance and results could differ using 25 VGUT3cKO mice instead of 19. In order to generate a proper comparison, authors must increase the number of VGUT3cKO mice.

Most of the average results are using Z-score, this data normalization reflects the number of standard deviation that values have with respect to the mean. Usually, the range of values is in between \pm few standards deviation (+3 to -3, +5 to -5...). However, here there are some results with Z-score values around 60 or 20 (Fig. 1i; Fig. 2f; Fig. 3d...), which is unusual. In order to understand the results, I need to see these graphs as absolute values (not normalized), with their standard deviation values, I also need to know the recording time window used for averaging. In general, the statistics, data collection, and analysis section (in methods) must be improved. For instance, it is written: "Generation of heatmap plots, estimation of area under the curve (AUC), and height peak were respectively generated and extracted using OriginPro 2021 V9.8.0.200 (OriginLab)" -Please, explain how does OriginPro to calculate it? "The sample sizes used in our study are about the same or exceed those estimated by power analysis (power = 0.9, α = 0.05)" ...- Which test has been performed to calculate the sample size?

Please, improve statistics/analysis section including these and other relevant details for better understanding of the data analysis.

Minor comments:

As a general comment, I found reading somewhat uncomfortable, perhaps due to the often calls to principal figures intermingled with supplementary figures. I think that some of the supplementary figures contain very relevant results that can be included as a subpanels in the current principal figures or as a new principal figure. For instance, the results comparing the action of VAcHTcKO and VGLUT3cKO are showed in figures Supp. Fig.3 and Supp. Fig.5 respectively. Authors can put together as a principal figure these results, perhaps leaving the descriptions between sexes to the supplementary information. Also some panels of the Supp. Fig. 7 can be perfectly integrated in the Fig. 4. I really recommend a restructuring of the principal figures.

The discussion is focus in impact that ACh and the DA has on the striatal microcircuits. However, in addition with the action of the DA or ACh on the direct activation of dopaminergic, nicotinic or muscarinic receptors, it is well known that ChI inhibit SPNs by disynaptic inhibition, mediated by others striatal GABAergic interneurons. Authors should discuss how the activation of some GABAergic interneurons can modulate the SPNs activity after ACh release.

Line 231... "Interestingly, VAcHTcKOs spent more time approaching the CS+ compared to the CS- during late reversal sessions" ... -Please, point out the number of figure showing this result.

Line 257... "We also observed a phasic ACh response during CS+ offset, but not during CS- offset that was significantly impaired in VAcHTcKOs (Supp.Fig.4g)" ... -Please check the figure, I only found the CS+ offset result.

Line 348... "Regarding D1-SPN activity, we first observed a generalized phasic increase during the onset of both the CS+ and the CS- in control and VAcHTcKO mice across all training sessions ($p > 0.05$) (Supp.Fig.7e)" ...-Please, check the figure and text.

Line 352...."Finally, following reward delivery the activity in control mice was characterized by a bi-phasic burst (Supp.Fig.7f)" ...-Please, check the figure and text.

Figure 5d will be more informative including the controls, together with the sham and rescued groups. Also check the figure alignment with respect to the left margin.

Line 606... "Animals not able to reach criteria after four consecutive pre-training sessions were excluded from the study". -It is also informative to know how many animals were excluded from the study.

Figure Supp 2b. Looking this figure is easy to see the serious lesion induced by the probe. Basically it removes the lateral ventricle, the secondary motor cortex, the cingulate cortex and the septal nuclei of one hemisphere. Did authors found changes in mice behavior after the probe implantation?

Reviewer #2 (Remarks to the Author):

This manuscript by Skirzewski and colleagues focuses on the cholinergic-dopaminergic interaction in the NAc, and the role of acetylcholine and glutamate release from striatal cholinergic interneurons in a Pavlovian task. First, the authors designed and validated a touchscreen-based autoshaping task. They further selectively lesioned ACh release or glutamate release from cholinergic interneurons and found ACh but Glutamate release from cholinergic interneurons distorted the learning process. They also showed that lesioned ACh release changed calcium signals of D1 and D2 neurons. Finally, the authors showed that re-expressing VAcHT in the NAc can restore ACh pauses and approach behaviours in VAcHTcKO mice. This study addressed the vital question of whether glutamate-corelease of cholinergic interneurons is as critical as ACh release in a Pavlovian task. However, some concerns can significantly undermine the conclusion authors made.

1. A long CS signal in a Pavlovian task is not uncommon. However, could authors also address why animals have similar initial responses (about 1s long?) to both CS+ and CS- (Fig1)?
2. It is unclear why the authors used z-score $([\Delta F/F - (\mu \text{ baseline})]/\sigma \text{ baseline})$, but not the $\Delta F/F$. Both dopamine neurons and cholinergic interneurons are tonically active. Baseline activity could also be critical to understanding how these two neurotransmitter systems work. It is also confusing why $\sigma \text{ baseline}$ was involved. Could the authors please give a detailed explanation?
3. Please give more details on how selective knockout of the vesicular ACh transporter (VAcHT, Fig.2a and Supp.Fig3-4) or the vesicular glutamate transporter (VGLUT3, Supp.Fig.5) was done. How was these knockout limited to the striatum but not the other brain areas? I have tried to understand this more with the paper published by the same group, but it is not very clear to me.
4. It seems that VAcHTcKO animals have a better ACh release than their controls at the beginning of CS? Please explain more about this. In addition, please show evidence of lesioned ACh release in the striatum.
5. To record from both D1 and D2 simultaneously is indeed a great idea. However, non-D2 neurons are not D1 neurons, as the authors also claimed in the manuscript. I agree that about 95% of neurons in the striatum are SPNs. However, they are also maybe the most silent ones. The interneurons (5% of striatal neurons) labelled with the Cre-Off AAV expressing GCaMP6s may contribute more fluorescence signals than D1 SPNs. The results would be more convincing if D1 neurons could be specifically labelled. Unfortunately, it would be very hard to do with the animals used in this study.
6. It is nice that the authors inserted VAcHT to adult VAcHTcKO in NAc to restore ACh release. However, comparing the ACh release level to the control animals is critical. Although the animals regained learning ability, it doesn't mean they use the same cellular mechanism as controls.

7. It would be good also to compare the 'rescued' learning process to the control animals.

Minor:

Please label the figure better. What signal was recorded is not always clear (e.g. fig2.c,d).

Reviewer #3 (Remarks to the Author):

Skirzewski and colleagues investigate the role of acetylcholine signaling in the nucleus accumbens in the development of Pavlovian conditioned approach behavior, dopamine signaling, and D1 and D2 neuron activity. Across a number of studies, they demonstrate that normal acetylcholine signaling from cholinergic interneurons is necessary for the development of cue-reward association underlying Pavlovian approach. Dopamine dynamics and D1 and D2-neuron activity in during Pavlovian learning are all disrupted when acetylcholine signaling is blunted via knock out of the vesicular acetylcholine transporter Overall these studies demonstrate that cholinergic interneurons, via acetylcholine, are critical regulators of striatal activity and play a central role in reward learning. I really liked this paper. The studies are thorough and rigorous. As a scholarly work it is also impressive, with thorough citing of the large relevant set of earlier studies. I like the combination of genetic and optical tools, and validation of the sensors. The results are timely, as there is growing appreciation for the role of CINs in striatal microcircuit function and motivated behavior. I have no major comments.

REVIEWER COMMENTS

Reviewer #1 (Remarks to the Author):

This interesting study demonstrates the close interaction between the striatal cholinergic interneurons and dopamine release in the mouse nucleus accumbens, during a autoshaping task performance of Pavlovian paradigm. Even the hypothesis of the DA-ACh relation is not new, it is very remarkable that this investigation dissects the action of acetylcholine and glutamate along the different phases of a Pavlovian task, both of them neurotransmitters that are coreleased by ChIs in the nucleus accumbens. To measure the DA or ACh release authors used genetically-encoded sensors and fibre photometry experiments together with transgenic mice. In addition, by means of calcium indicators, authors measured the activity of D1- and D2- SPNs during the task performance.

A major concern is related with the measure of the D1- and D2- SPNs activity in both VChTcKo and control mice. In a recent article, published as preprint, it was compared the activity of striatal SPNs recorded by calcium imaging using GCaMP6s (the same as the one used in this article) with respect to the action potentials recorded extracellularly by single unit in the same neurons (Legaria et al., 2021). Basically, they found that calcium activity does not correlate with the somatic action potentials on SPNs. Instead, calcium activity correlates with changes on fluorescence in the neuropil, reflecting the inputs that neurons receive, instead than the outputs (the action potentials). Thus, authors must demonstrate that the fluorescence changes recorded in SPN are directly reflecting action potentials. If finally, the calcium signal is related to the SPN inputs instead to their action potentials, authors must reinterpret their results.

Response: *We thank the reviewer for this important comment. We too became aware of this study shortly after submission of our manuscript. Although until very recently it was a non-peer-reviewed preprint, it has now been published in Nature Neuroscience. We agree it is an important and comprehensive study, carried out by a highly competent team of researchers. Thus, we agree with the reviewer that their findings are almost certainly correct and there is no reason to expect that were we to re-do their experiments, that we would find anything substantially different. Furthermore, such experiments are dauntingly complex, and would ideally require simultaneous electrophysiological recordings from VChTcKO and control littermate mice, which would be prohibitively challenging! Therefore, we agree with the reviewer that reinterpretation is required. Specifically, following the suggestion of Legaria et al. (2022), we interpret the observed SPN calcium signals, regulated by continuous updating of ACh and DA, as likely reflecting a dendritic 'eligibility trace', that leads to weight change in a susceptible synapse only if an additional neuromodulatory signal (i.e., DA, ACh) is present, related to novelty, punishment, or reward events (Gerstner et al. 2018).*

We have added consideration of the reviewers' observations of Legaria et al. to the manuscript in the following ways:

(lines 1-2): "Continuous Cholinergic-Dopaminergic Updating in the Nucleus Accumbens Underlies Approaches to Reward-Predicting Cues"

(lines 121-122): "...we used genetically-encoded sensors and fibre photometry to record millisecond dynamics of ACh, DA, and calcium in putative D1- and D2-SPNs,..."

(Lines 318-319) “Dysfunctional cholinergic signalling in the striatum drives abnormal direct and indirect spiny projecting neuron calcium dynamics”

(lines 328-331) “...to understand the association between NAc ACh, DA, SPNs, and behaviour, we simultaneously studied the calcium activity of putative D1- and D2-SPNs during the acquisition of cue-motivated approach behaviours in the Autoshaping task, in both VAcTcKO and control mice.”

(line 415) “...coordinate circuit mechanisms regulating the calcium activity of the direct and indirect...”

(lines 533-550): “It is important to highlight that recent elegant work from Legaria et al.⁹³ demonstrated that the striatal SPN calcium fibre photometry signal may reflect not only spiking dynamics, and instead much of the signal may arise from the dense dendritic arborization of neurons. Therefore, following the suggestion of Legaria et al. (2022), we interpret the observed SPN calcium signals, regulated by continuous updating of ACh and DA, as possibly reflecting a dendritic ‘eligibility trace’. The eligibility trace is posited by emerging theories of synaptic plasticity as a kind of flag, set at the synapse by the co-activation of pre- and postsynaptic neurons, that leads to weight change in a susceptible synapse only if an additional factor such as novelty, punishment, or reward is present¹²³. Moreover, this additional factor is often implemented by the phasic activity of neuromodulators such as DA and ACh¹²⁴⁻¹²⁶. Therefore, our observations may reflect an eligibility trace for synaptic plasticity and behavioural conditioning^{124, 125, 127}, mediated by a continuous updating of ACh and DA dynamics, and triggered by behaviourally relevant stimuli. Future work is needed to address how the heterogeneous contribution of dopaminergic and muscarinic receptors expressed in D1- and D2-SPNs may regulate the shape, volume, and stability of dendritic spines¹²⁸, and how this could influence changes in synaptic plasticity mechanisms regulating behaviour.”

(lines 573-576): “Taken together, our work suggests that an intricate balance between DA-ACh, and its reciprocal rapid updating during learning, is critical for the regulation of local network mechanisms and neuronal engrams underlying approaches to reward-predicting cues.”

Another important concern is related with number of animals that authors used to demonstrate that ACh but not glutamate is necessary for the acquisition of the task. This conclusion is based on the independent comparisons between VAcTcKO vs control mice and the VGLUT3cKO vs control mice. To that end, authors used 25 VAcTcKO and 19 VGUT3cKO, reaching the statistical significance in the first group. Samples are unbalance and results could differ using 25 VGUT3cKO mice instead of 19. In order to generate a proper comparison, authors must increase the number of VGUT3cKO mice.

Response: *Because the use of genetically modified mutant mice generated in our mouse colony relies on the constant supply of new pups from breeders, it is impossible to predict the specific number of mice that a researcher will obtain to perform an experiment. Instead, we estimate the minimum number of mice required to get statistical significance with a priori power analysis. Our prior touchscreen studies yielded effect sizes (Cohen’s f) ranging from 0.41 to 0.79 for two-way ANOVA and repeated measures analyses. Using the effect size of 0.41, we computed that sample sizes of at least 16 mice would achieve 80% power, in line with our prior work for touchscreen experiments. Hence, even with 19 mice, we would have enough statistical power for these experiments.*

These are the main reasons we have discrepancies in the total number of animals used in this study, including number of offspring, loss of animals due to surgery, etc. Nonetheless, as requested we added 4 VGLUT3cKO mice to the study. Additionally, we also added 4 control VChTcKO mice as the sample size of this group was also unbalanced. Our new updated data with balanced sample size in all groups is as follows: VGLUT3cKO N=23, control VGLUT3cKO N=24; VChTcKO N=25, control VChTcKO N=24.

Please see updated figures 2c, 2d, and supplementary figures 3 and 4.

Most of the average results are using Z-score, this data normalization reflects the number of standard deviation that values have with respect to the mean. Usually, the range of values is in between \pm few standards deviation (+3 to -3, +5 to -5...). However, here there are some results with Z-score values around 60 or 20 (Fig. 1i; Fig. 2f; Fig. 3d...), which is unusual. In order to understand the results, I need to see these graphs as absolute values (not normalized), with their standard deviation values, I also need to know the recording time window used for averaging.

Response: *As per the request of the referee, please see $\Delta F/F$ values of updated Fig.1f, Fig.2f, Supp.Fig.2d, 2g-h, Supp.Fig.5g, Supp.Fig.7a-b, and Supp.Fig.8b-c. The mean signal during CS presentation was estimated by averaging the entire 10 seconds of either CS+ or CS- presentation. The AUC signal (and height peak) during reward delivery/collection was obtained by generating a ROI of approximately 5 seconds after CS+ offset for dopamine, and of ~10 seconds for ACh. Analysis ROI window to estimate amplitude of CS onset and offset was of ~2 seconds duration.*

In general, the statistics, data collection, and analysis section (in methods) must be improved. For instance, it is written: "Generation of heatmap plots, estimation of area under the curve (AUC), and height peak were respectively generated and extracted using OriginPro 2021 V9.8.0.200 (OriginLab)".... - Please, explain how does OriginPro to calculate it? "The sample sizes used in our study are about the same or exceed those estimated by power analysis (power = 0.9, α = 0.05)"...- Which test has been performed to calculate the sample size? Please, improve statistics/analysis section including these and other relevant details for better understanding of the data analysis.

Response: *We calculated the effect sizes with a priori partial eta squared (η_p^2) power analysis (power =0.9, α =0.05, 95% confidence intervals) for repeated measures two-way ANOVA (Lakens, 2013), to estimate sample sizes in our touchscreen Autoshaping behavioural analyses. Our touchscreen studies yielded effect sizes (Cohen's f) ranging from $\eta_p^2= 0.41$ to 0.75. Using the effect size of 0.41, we computed that sample sizes of at least 12 mice would achieve 90% power, indicating sufficient statistical power for these experiments. We have expanded the method section to help clarify how generation of heat maps, estimation of area under the curve and height peaks, and power analysis were estimated (lines 830-869). We also reported power analysis results in lines 1629-1633.*

(lines 830-869): "Behavioural data was extracted from ABET II Video Touch software V21.02.26 (Lafayette Instruments). The data supporting these findings can be visualized and are freely available at MouseBytes (<https://mousebytes.ca/home>). Generation of heatmaps, estimation of area under the curve (AUC), and height peak analysis of events were obtained using OriginPro 2021 V9.8.0.200 (OriginLab Corporation, Northampton, MA). Briefly, heatmaps illustrating $\Delta F/F$ or z-score from GRABDA, ACh3.0 or

calcium signaling consisted on individual trials, or averaged trials (20 trials/session). Each began with 1s long baseline before CS onset, followed by 10s long CS+ or CS- presentation followed by the delivery of a single reward (7 μ l) after CS+ offset. DA and calcium signal during CS presentation was calculated by averaging the signal during CS presentation. To find peaks, calculate height, and to integrate the AUC, the averaged trial signal from each session was used to find and calculate peaks (curve) within the ROI, obtaining the baseline by estimation of end points weighted (15%) within the window search. Alternatively, for the estimation of number of peaks during validation experiments recording calcium (Supp.Fig8d), the 1st derivative signal within the 10 min window search (saline vs. cocaine) was applied followed by the Savitzky-Golay method to smooth the signal. To calculate the AUC and height peak of the DA signal after reward delivery, the window search was set for the following 5s after CS+ offset. The AUC and height of the peak of the reward response during ACh recordings were estimated by setting the window search for the 10s after CS+ offset. Alternatively, the mean signal post-reward delivery for the calcium recordings in D1-SPN and D2-SPN was calculated by averaging the signal during the 10s immediately post CS offset. All data were imported to Graph Prism V9.3.1 for Windows 10 (GraphPad Software, San Diego, CA) for statistical analysis. Initially, a normality test (D'Agostino-Pearson) was performed to determine whether parametric or non-parametric tests were appropriate. No assumptions or corrections were made prior to data analysis. Differences between two groups were always examined using a two-tailed Student's t-test, where $p < 0.05$ was considered significant and $p > 0.05$ was considered non-significant. Comparisons between multiple groups were performed using analysis of variance (ANOVA; one-way and two-way with repeated measures), followed by Sidak's multiple comparisons test. Alternatively, when a dataset had missing values, we compared groups using a linear mixed-effects ANOVA model. A simple linear regression analysis was used to correlate DA and ACh signaling with approaches to the CS presentation. Estimation of sample size in Autoshaping task experiments was estimated using a partial eta squared (η^2) power analysis for repeated measures two-way ANOVA (power=0.9, $\alpha=0.05$)¹³⁹. For experiments combining fibre photometry and touchscreens, we used standard sample sizes ($N=7-11$) as previously reported^{87, 93}. All cell counting, ISH, and immunohistochemistry experiments were performed by an experimenter blind to the experimental condition. Estimation of sample size was estimated based on previous reports. All data were expressed as mean \pm s.e.m. and $p < 0.05$ were considered as statistically significant."

(lines 1629-1633): "...(two-way RM ANOVA session x CS interaction, Acquisition females: $F(9,198)=4.809$, $p < 0.0001$, $\eta_p^2=0.41$ [0.28, 0.53]; Reversal females: $F(9,198)=14.12$, $p < 0.0001$, $\eta_p^2=0.75$ [0.62, 0.89]; Acquisition males: $F(9,198)=6.378$, $p < 0.0001$, $\eta_p^2=0.48$ [0.35, 0.61]; Reversal males: $F(9,198)=10.42$, $p < 0.0001$, $\eta_p^2=0.64$ [0.51, 0.77])."

Minor comments:

As a general comment, I found reading somewhat uncomfortable, perhaps due to the often calls to principal figures intermingled with supplementary figures. I think that some of the supplementary figures contain very relevant results that can be included as a subpanels in the current principal figures or as a new principal figure. For instance, the results comparing the action of VACHTcKO and VGLUT3cKO are showed in figures Supp. Fig.3 and Supp. Fig.5 respectively. Authors can put together as a principal figure these results, perhaps leaving the descriptions between sexes to the supplementary information. Also some panels of the Supp. Fig. 7 can be perfectly integrated in the Fig. 4. I really recommend a restructuring of the principal figures.

Response: We thank the reviewer for these suggestions to improve the manuscript flow. We included results comparing VACHTcKO and VGLUTcKO mice, and their corresponding control littermates in updated Figure 2. For consistency, we also included RNAscope analysis of VGLUT3cKO mice originally in Supp.Fig5a into Fig.2b and moved panels from Supp.Fig.7 (now Supp.Fig.8) into updated Fig.4b, 4c, 4g, and 4i. Overall, we have edited all the figures and figure legends of this manuscript to better help the reader.

The discussion is focus in impact that ACh and the DA has on the striatal microcircuits. However, in addition with the action of the DA or ACh on the direct activation of dopaminergic, nicotinic or muscarinic receptors, it is well known that Chl inhibit SPNs by disynaptic inhibition, mediated by others striatal GABAergic interneurons. Authors should discuss how the activation of some GABAergic interneurons can modulate the SPNs activity after ACh release.

Response: We thank the reviewers for this suggestion, we have included a paragraph within the discussion section (lines 89-93, 434-436) commenting this important mechanism.

(lines 89-93): “CINs also directly regulate, via M1- and M2-class muscarinic receptors²²⁻²⁶, or indirectly via the activation of parvalbumin-positive GABAergic interneurons^{26, 27}, the concurrent activity of D1- and D2-expressing spiny projecting neurons (SPNs) projecting to the direct striatonigral (D1-SPNs) or indirect striatopallidal (D2-SPN) basal ganglia pathways²⁸.”

(lines 434-436): “Moreover, striatal GABAergic interneurons appear to play a fundamental role in the modulation of the network SPN activity mediated by CIN-dependent disynaptic inhibitory mechanisms^{26, 27}.”

Line 231... “Interestingly, VACHTcKOs spent more time approaching the CS+ compared to the CS- during late reversal sessions” ... -Please, point out the number of figure showing this result.

Response: We refer to Fig.2c and Supp.Fig.3b. We have edited the referred paragraph for clarity.

(lines 226-233): “We found that VACHTcKO (VACHTcKO: N=25, n=11♂, n=14♀; control: N=24, n=13♂, n=11♀; Fig.2c and Supp.Fig.3), but not VGLUT3cKO mice (VGLUT3cKO: N=23, n=12♂, n=11♀; control: N=24, n=12♂, n=12♀; Fig.2d and Supp.Fig.4) failed to discriminate between the CS+ and CS- during acquisition sessions, demonstrated by their equal time spent approaching the CS+ and CS- during presentation (Fig.2c and Supp.Fig.3b). Interestingly, VACHTcKOs spent more time approaching the CS+ compared to the CS- during late reversal sessions, suggesting that some basic learning ability is preserved.”

Line 257... “We also observed a phasic ACh response during CS+ offset, but not during CS- offset that was significantly impaired in VACHTcKOs (Supp.Fig.4g)” ... -Please check the figure, I only found the CS+ offset result.

Response: *Apologies for the misunderstanding. The paragraph was meant to indicate that a phasic ACh response was only observed during CS+ offset. This event was absent during CS- offset, and therefore no quantification was done. We have edited the referred paragraph to clarify (lines 257-259).*

(lines 257-259): “We also observed a phasic ACh response during CS+ offset that was significantly impaired in VChTcKO mice (Supp.Fig.5i). This event was not observed during CS- offset.”

Line 348... “Regarding D1-SPN activity, we first observed a generalized phasic increase during the onset of both the CS+ and the CS- in control and VChTcKO mice across all training sessions ($p > 0.05$) (Supp.Fig.7e)” ...-Please, check the figure and text.

Response: *We have edited the paragraph for accuracy (lines 356-361).*

(lines 356-361): “Regarding D1-SPNs, we first observed during the CS+ and CS- onset a phasic calcium increase across all training sessions in both control and VChTcKO mice (Fig.4g). In VChTcKO mice, this event was significantly larger during the first two acquisition sessions in the CS+ compared to the CS-. Second, the calcium signal amplitude significantly reduced as approaches toward the CS+ increased in controls but not in VChTcKO mice (Fig.4h, top panels).”

Line 352....”Finally, following reward delivery the activity in control mice was characterized by a bi-phasic burst (Supp.Fig.7f)” ...-Please, check the figure and text.

Response: *We have edited the paragraph for accuracy (lines 361-364).*

(lines 361-364): “Finally, following reward delivery the calcium signal in control mice was characterized by a bi-phasic burst (Fig.4i) and pause event (Fig.4j, top-left panel) across acquisition and reversal sessions. The amplitude of the phasic (burst) calcium increase was larger in control than VChTcKO mice.”

Figure 5d will be more informative including the controls, together with the sham and rescued groups. Also check the figure alignment with respect to the left margin.

Response: *We have included additional plot of control littermate mice from experiments presented in Fig.2 (mice performing the Autoshaping task while recording ACh in NAc). We have checked figure alignment to left margin. Check updated Fig.5e.*

Line 606... “Animals not able to reach criteria after four consecutive pre-training sessions were excluded from the study”. -It is also informative to know how many animals were excluded from the study.

Response: *We have included the total of mice that did not reach pre-training criteria in the methods section (see lines 658-661).*

(lines 658-661): “Animals not able to reach criteria after four consecutive pre-training sessions were excluded from the study. In total, 3 VGLUT3cKO (n=2♂, n=1♀), 1♀ control VGLUT3, and 1♂ VChTcKO were excluded.”

Figure Supp 2b. Looking this figure is easy to see the serious lesion induced by the probe. Basically it removes the lateral ventricle, the secondary motor cortex, the cingulate cortex and the septal nuclei of one hemisphere. Did authors found changes in mice behavior after the probe implantation?

Response: *We are aware that implanting a 400 µm diameter fiber optic does cause a considerable brain lesion. However, no behavioural differences were observed across sexes and genotypes between mice undergoing surgeries vs. no surgeries. To help clarify that mice undergoing surgeries (Fig.1d, Fig.3d, Fig.4c, Fig.5e, and Supp.Fig.5d) behave similarly to naïve mice (Fig.2c, 2d, Supp.Fig.1, Supp.Fig.3, Supp.Fig.4, and Supp.Fig.6), we have included a paragraph in the result section (see lines 180-183).*

(lines 180-183): “We found that mice tethered for fibre photometry recordings behaved similarly to control mice without fibre optical implants during the Autoshaping task ($p>0.05$, Fig.1d and Supp.Fig.1a), indicating no major effect of tethering or surgical implants.”

Reviewer #2 (Remarks to the Author):

This manuscript by Skirzewski and colleagues focuses on the cholinergic-dopaminergic interaction in the NAc, and the role of acetylcholine and glutamate release from striatal cholinergic interneurons in a Pavlovian task. First, the authors designed and validated a touchscreen-based autoshaping task. They further selectively lesioned ACh release or glutamate release from cholinergic interneurons and found ACh but Glutamate release from cholinergic interneurons distorted the learning process. They also showed that lesioned ACh release changed calcium signals of D1 and D2 neurons. Finally, the authors showed that re-expressing VAcHT in the NAc can restore ACh pauses and approach behaviours in VAcHTcKO mice. This study addressed the vital question of whether glutamate-corelease of cholinergic interneurons is as critical as ACh release in a Pavlovian task. However, some concerns can significantly undermine the conclusion authors made.

1. A long CS signal in a Pavlovian task is not uncommon. However, could authors also address why animals have similar initial responses (about 1s long?) to both CS+ and CS- (Fig1)?

Response: *Several lines of evidence suggest that in addition to the reward prediction error, dopamine activity also encodes arousal sensory and novelty information unrelated to reward (Strecker and Jacobs, 1985; Ljungberg et al. 1992; Menegas et al, 2017). It is possible that the transient phasic DA response during CS onset indicates a mechanism of attention to stimuli, regardless of the associated contingency of the task. Perhaps more likely, seminal work has previously suggested that similar conditioned stimuli (such as CS+ and CS-) can be briefly generalised by rodents during their onset, due to initial activation of ‘low-resolution’ circuit pathways that do not discriminate the contingency of the stimulus presented, but instead its arrival. We have added a paragraph on the discussion section to address this comment.*

(lines 486-497). “Finally, we observed a ~1s long DA response at the onset of the CS presentation, even to the CS- (Fig.1h-i). Several lines of evidence suggest that in addition to the reward prediction error, DA responses are also observed during arousing sensory and/or novel events¹¹⁰⁻¹¹³. However perhaps more likely is that this is a conditioned response that briefly generalises. Both CS+ and CS- are similar stimuli (large bright rectangles) that differ only in their spatial location. It is perhaps not surprising that following conditioning, when a large bright stimulus appears, there is a generalised DA response even to

the CS-, which is rapidly curtailed once the system identifies the stimulus as the CS-. This explanation is similar to the idea that two sensory systems pass information to reward circuitry: a ‘low road’, which provides rapid but low-resolution information, and a ‘high road’ that provides high resolution information that becomes available following a brief delay¹¹⁴⁻¹¹⁶.”

2. It is unclear why the authors used z-score ($(\Delta F/F - (\mu \text{ baseline}))/\sigma \text{ baseline}$), but not the $\Delta F/F$. Both dopamine neurons and cholinergic interneurons are tonically active. Baseline activity could also be critical to understanding how these two neurotransmitter systems work. It is also confusing why $\sigma \text{ baseline}$ was involved. Could the authors please give a detailed explanation?

Response: *The z-score is useful to standardize $\Delta F/F$ data across animals to correct for the decay in the fluorescence signal over time (photobleaching), as well as for differences in baseline fluorescence related to biosensor expression level and proximity of fiber probe to biosensor (please see Fig.1f, Fig.2f, Supp.Fig.2d and Supp.Fig.8b, 8c to illustrate variability of raw $\Delta F/F$ baseline signal). Moreover, we used the pre-CS onset baseline period (1 second long) as the input values to calculate z-scores (often called ‘baseline z-score’). This approach is useful to reveal changes in fluorescence dynamics (i.e. DA, ACh, and calcium) in part because baseline periods tend to have relatively low variability. As such, these results in a time series are interpreted in terms of standard deviations and mean during the baseline period. Baseline z-scores are conceptually justifiable because the z-score is then the number of standard deviations from the mean when a subject is waiting to initiate as new trial (at rest).*

Finally, it is important to highlight that fibre photometry is not suitable to compare tonic baseline activity (due to levels of expression of the sensors, fibre positing etc.) but rather it provides insight in variations of time-series data with high temporal resolution. To this end, Favier et al. 2020 previously reported that VAcHTcKO mice show a significant reduction in extracellular baseline ACh levels and DA-evoked events in the striatum, combining the use of microdialysis and voltametric recordings. We have cited this work several times across the manuscript to help clarify on this comment.

(lines 250-255): “A previous report using microdialysis has shown that tonic striatal extracellular ACh levels in VAcHTcKO are significantly reduced (~95%)⁴⁹, which may limit the ability to detect decreased cholinergic signals using ACh3.0. It is therefore likely that cholinergic tone in VAcHTcKO mice is so low that changes in CIN activity (such as pauses in activity) are unable to further modulate cholinergic tone.”

(Lines 455-458): “Reduced VAcHT expression decreases ACh release to levels below the detection limit of microdialysis⁴⁹. Because striatal baseline ACh levels are very low in VAcHTcKO, CIN pauses in these mutants likely lack the potential to neuromodulate local plasticity mechanisms within the NAc.”

3. Please give more details on how selective knockout of the vesicular ACh transporter (VAcHT, Fig.2a and Supp.Fig3-4) or the vesicular glutamate transporter (VGLUT3, Supp.Fig.5) was done. How was these knockout limited to the striatum but not the other brain areas? I have tried to understand this more with the paper published by the same group, but it is not very clear to me.

Response: *Striatal cholinergic interneurons simultaneously co-express VAcHT, VGLUT3 and D2 receptors. We used genetically modified mice in which striatal VAcHT or VGLUT3 was selectively eliminated from*

cholinergic interneurons using the Cre-Lox recombination approach, by crossing D2-Cre mice with VAcHT-flox or VGLUT3-flox mice. A full description of the generation and molecular characterization of selective VAcHT and VGLUT3 KO mice is found in a recent published work (Favier et al. 2020, <https://pubmed.ncbi.nlm.nih.gov/33164988/>). We elected to use D2-Cre so we could use the same Cre lines to delete VAcHT and VGLUT3. One alternative would be to use a different Cre mouse (ChAT-Cre and VGlut3-Cre). However, these lines would not be selective and they would generate full KOs of VAcHT or VGLUT3. We are aware of ectopic expression of Cre in any line used (Luo et al. 2020, Neuron), hence the D2-Cre line offers the best compromise to use the same line to delete both genes. To make sure that the effects we observe are indeed due to NA VAcHT, we performed a rescue experiment with a AAV-VAcHT to rescue the phenotypes. We included a paragraph in the Methods section (Lines 594-602) to better clarify this point.

(Lines 594-602): “Striatal cholinergic interneurons simultaneously co-express VAcHT, VGLUT3 and D2 receptors⁷⁴. We used genetically modified mice in which striatal VAcHT or VGLUT3 was selectively eliminated from cholinergic interneurons using the Cre-Lox recombination approach as previously described⁴⁹. We elected to use D2-Cre so we could use the same Cre line to delete VAcHT and VGLUT3. One alternative would be to use a different Cre driver mouse line (i.e. ChAT-Cre and VGlut3-Cre). However, these lines would not be selective, and they would generate full KOs of VAcHT or VGLUT3. Moreover, we are aware of ectopic expression of Cre in any line used¹³⁵, hence the D2-Cre line offers the best compromise to use the same line to delete both genes.”

4. It seems that VAcHTcKO animals have a better ACh release than their controls at the beginning of CS? Please explain more about this. In addition, please show evidence of lesioned ACh release in the striatum.

Response: As discussed in lines 455-458, Favier et al. 2020 demonstrated that VAcHTcKO mice have a dramatic reduction (~95%) of extracellular ACh levels in the striatum using microdialysis. Given this ACh deficit, it is likely that the signal-to-noise fluorescence ratio obtained from the GRAB-ACh sensor is significantly reduced, and therefore generating noisier ACh dynamics. Moreover, because the ACh signal is expressed as z-score, the differential between baseline vs. time-locked events seems larger. Finally, we must consider that VAcHTcKO mice still receive cholinergic innervation from LDT/PPTg that could partially contribute with some of the signal fluorescence from GRAB-ACh. It is unclear how the lack of ACh release from CINs affects the local release from cholinergic LDT/PPTg projecting neurons, despite the fact they don't seem to contribute to behavioural deficits in the Autoshaping task (Supp.Fig.6). We added in Fig.2f a representative $\Delta F/F$ trace to help clarify this point.

(lines 455-458): “Reduced VAcHT expression decreases ACh release to levels below the detection limit of microdialysis⁴⁹. Because striatal baseline ACh levels are very low in VAcHTcKO, CIN pauses in these mutants likely lack the potential to neuromodulate local plasticity mechanisms within the NAc.”

(lines 460-463): “Conversely, we suggest that the anomalous endophenotype in VAcHTcKO mice results from disturbed ACh storage from cholinergic synapses leading to a blunted vesicular ACh release from CINs^{76, 106, 107}, that reduces ACh signal-to-noise ratio.”

5. To record from both D1 and D2 simultaneously is indeed a great idea. However, non-D2 neurons are not D1 neurons, as the authors also claimed in the manuscript. I agree that about 95% of neurons in the striatum are SPNs. However, they are also maybe the most silent ones. The interneurons (5% of striatal neurons) labelled with the Cre-Off AAV expressing GCaMP6s may contribute more fluorescence signals than D1 SPNs. The results would be more convincing if D1 neurons could be specifically labelled. Unfortunately, it would be very hard to do with the animals used in this study.

Response: *We agree our experimental design cannot rule out that some of the fluorescence signaling collected from GCaMP6s.Cre-Off can arise from neurons other than D1-SPNs (i.e., inhibitory GABAergic interneurons). Same is true for the fluorescence signal from jRCaMP1a.Cre-On, as in addition to D2-SPNs, cholinergic interneurons also express D2 receptors. However, previous work by Markowitz et al. 2018 and Meng et al. 2018 demonstrated little interneuron fluorescence contamination in the striatum when combining the use of the Cre-On/Cre-Off system and multielectrode recordings in behaving D1-Cre, D2-Cre, and A2A-Cre mice. In our hands, it seems unlikely that the observed calcium dynamics arising from GCaMP6a and jRCaMP1a are the result of fluorescence contamination from interneurons because 1) our recorded events are time-locked to task contingencies, coinciding with elevated SPN activity necessary to encode information (Kreitzer, 2009, Kravitz and Kreitzer 2012, Markowitz et al. 2018, Cox and Witten 2019), 2) the observed in vivo dynamics between jRCaMP1a.DIO (Fig.4d-e) and ACh3.0 (Fig.2f-h) seem to antagonize after reward delivery (increased jRCaMP1a but decreased acetylcholine), despite both population of neurons (D2-SPNs and cholinergic interneurons) expressing D2 receptors, and 3) our validation experiments recording calcium dynamics during systemic cocaine administration (Supp.Fig.8d) are consistent with previous reports (i.e. Calipari et al. 2016) suggesting changes in the activity of the direct and indirect SPN pathways.*

In any case, we have clarified the fluorescence signal from GCaMP6s and jRCaMP1a is equivalent to the activity of 'putative' D1- and D2-SPNs, respectively (see lines 328-331). Also, is important to consider that recent work (Legaria et al. 2022) has demonstrated that calcium recordings from SPN using fibre photometry arise from neuropil (see response to reviewer#1). We have included additional discussion and re-interpretation of our findings to properly address this new evidence.

(lines 328-331): "...to understand the association between NAc ACh, DA, SPNs, and behaviour, we simultaneously studied the calcium activity of putative D1- and D2-SPNs during the acquisition of cue-motivated approach behaviours in the Autoshaping task, in both VAcHTcKO and control mice."

(lines 343-346): "Finally, considering recent observations by Legaria et al.⁹³ indicating that calcium dynamics recorded from striatal SPNs may not reflect spiking-related events but instead may be non-somatic (dendritic) changes, we interpreted our calcium recordings as likely arising from dendritic neuronal sub-structures."

6. It is nice that the authors inserted VAcHT to adult VAcHTcKO in NAc to restore ACh release. However, comparing the ACh release level to the control animals is critical. Although the animals regained learning ability, it doesn't mean they use the same cellular mechanism as controls.

Response: *Our fiber photometry experiment using ACh3.0 is a valuable tool to analyze in vivo time locked ACh dynamics. Unfortunately, it is not possible to correlate fluorescence signal with changes in baseline release. Thus, although our study does not directly demonstrate that inserting VAcHT into the NAc of*

VACHTcKO mice increases extracellular ACh baseline levels, it is important to highlight that in the striatum, only cholinergic interneurons require VACHT to transfer ACh from the cytoplasm into synaptic vesicles. Currently, there is no other function for VACHT that we are aware.

It is unclear how the expression of VACHT in cholinergic interneurons or any other neuronal type within the NAc may mediate a parallel cellular mechanism capable to restore the ability of KO mice to learn approach behaviors during the Autoshaping task. Moreover, our observation that VACHT-‘rescued’ VACHTcKO mice can regain the potential to generate ACh ‘pauses’ during reward delivery, in parallel with restored learning ability, provides strong evidence for the ACh pause mechanism in the processing of incentive salience, as previously demonstrated. We have added a paragraph within the manuscript (lines 569-573) to address this comment.

(lines 569-573): “Finally, although our study does not directly demonstrate that inserting VACHT into the NAc of VACHTcKO mice increases extracellular ACh baseline levels, it is important to highlight that in the striatum, only cholinergic interneurons require VACHT to transfer ACh from the cytoplasm into synaptic vesicles¹³⁰. Currently, there is no other function for VACHT that we are aware of.”

7. It would be good also to compare the ‘rescued’ learning process to the control animals.

Response: *We have included additional plot of control littermate mice from experiments presented in Fig.2 (mice performing the Autoshaping task while recording ACh in NAc) and corresponding analysis in Figure 5e.*

Minor:

Please label the figure better. What signal was recorded is not always clear (e.g. fig2.c,d).

Response: *We have edited and added additional labels across all figures to improve clarity.*

Reviewer #3 (Remarks to the Author):

Skirzewski and colleagues investigate the role of acetylcholine signaling in the nucleus accumbens in the development of Pavlovian conditioned approach behavior, dopamine signaling, and D1 and D2 neuron activity. Across a number of studies, they demonstrate that normal acetylcholine signaling from cholinergic interneurons is necessary for the development of cue-reward association underlying Pavlovian approach. Dopamine dynamics and D1 and D2-neuron activity in during Pavlovian learning are all disrupted when acetylcholine signaling is blunted via knock out of the vesicular acetylcholine transporter Overall these studies demonstrate that cholinergic interneurons, via acetylcholine, are critical regulators of striatal activity and play a central role in reward learning. I really liked this paper. The studies are thorough and rigorous. As a scholarly work it is also impressive, with thorough citing of the large relevant set of earlier studies. I like the combination of genetic and optical tools, and validation of the sensors. The results are timely, as there is growing appreciation for the role of CINs in striatal microcircuit function and motivated behavior. I have no major comments.

Response: *Thank you for your positive and encouraging feedback.*

REVIEWERS' COMMENTS

Reviewer #1 (Remarks to the Author):

This is a very interesting and rigorous study that helps to clarify the role of the cholinergic neurons in the ventral striatum. Miguel Skirzewski and colleagues did a serious effort, answering all my questions and suggestions, clarifying different parts of the manuscript and explaining its limitations. Thus, I have no major comments and I endorse its publication in Nature Communication.

Reviewer #2 (Remarks to the Author):

I don't have further concerns. Well done!

Reviewer #3 (Remarks to the Author):

This revised report from Skirzewski and colleagues follows up on a very strong initial submission. In the initial review, I had no major concerns, and continue to hold that view. This is a comprehensive, timely set of studies that nicely make use of a combination of genetic, imaging, and behavioral approaches to reconcile some longstanding hypotheses about the connection between acetylcholine and dopamine signaling in the striatum as it relates to reward learning.

I do not think that a revised manuscript must reconcile with the results of Legaria et al., 2021, which question the connection between fiber photometry recordings and individual neuron action potentials in the striatum. First, this is one report that has yet to be independently validated/replicated and so does not justify the complex experiments and cost required to answer the question posed by Reviewer 1 in the current paper. Second, the result is not essential to interpretations of the paper - calcium recordings using fiber photometry have never been suggested to reflect the activity of *individual neurons* and are not said to do so here. This wouldn't really be expected given the measurement methods anyway. Calcium signals do reflect neural activity and essential neural signaling processes, and so are still a valid measure, as they are in the 100s of other studies using these methods. Further, a large portion of the photometry data in the paper are using other biosensors that reflect specific neurotransmitter activity.

Use of Z scored delta F/F for the primary quantification of the photometry signals is standard practice for the trial-based analysis that is reported here.

REVIEWER COMMENTS

Reviewer #1 (Remarks to the Author):

This is a very interesting and rigorous study that helps to clarify the role of the cholinergic neurons in the ventral striatum. Miguel Skirzewski and colleagues did a serious effort, answering all my questions and suggestions, clarifying different parts of the manuscript and explaining its limitations. Thus, I have no major comments and I endorse its publication in Nature Communication.

Response: *Thank you for your positive feedback.*

Reviewer #2 (Remarks to the Author):

I don't have further concerns. Well done!

Response: *Thank you.*

Reviewer #3 (Remarks to the Author):

This revised report from Skirzewski and colleagues follows up on a very strong initial submission. In the initial review, I had no major concerns, and continue to hold that view. This is a comprehensive, timely set of studies that nicely make use of a combination of genetic, imaging, and behavioral approaches to reconcile some longstanding hypotheses about the connection between acetylcholine and dopamine signaling in the striatum as it relates to reward learning.

I do not think that a revised manuscript must reconcile with the results of Legaria et al., 2021, which question the connection between fiber photometry recordings and individual neuron action potentials in the striatum. First, this is one report that has yet to be independently validated/replicated and so does not justify the complex experiments and cost required to answer the question posed by Reviewer 1 in the current paper. Second, the result is not essential to interpretations of the paper - calcium recordings using fiber photometry have never been suggested to reflect the activity of *individual neurons* and are not said to do so here. This wouldn't really be expected given the measurement methods anyway. Calcium signals do reflect neural activity and essential neural signaling processes, and so are still a valid measure, as they are in the 100s of other studies using these methods. Further, a large portion of the photometry data in the paper are using other biosensors that reflect specific neurotransmitter activity.

Use of Z scored delta F/F for the primary quantification of the photometry signals is standard practice for the trial-based analysis that is reported here.

Response: *We again appreciate the positive and encouraging feedback regarding our manuscript. We agree with the reviewer's comments. We are happy with the revised version of this manuscript, and we agree there is no need for further editing.*